# WEAK-TO-STRONG GENERALIZATION THROUGH THE DATA-CENTRIC LENS

**Changho Shin, John Cooper, Frederic Sala**
Department of Computer Science
University of Wisconsin-Madison
`{cshin23,jfcooper2,fredsala}@wisc.edu`

## ABSTRACT

The weak-to-strong generalization phenomenon is the driver for important machine learning applications including highly data-efficient learning and, most recently, performing *superalignment*. While decades of research have resulted in numerous algorithms that produce strong empirical performance, understanding what **aspects of data** enable weak-to-strong generalization has been understudied. We propose a simple data-centric mechanism that characterizes weak-to-strong generalization: the *overlap density*. Intuitively, generalization tracks the number of points that contain overlaps, i.e., both easy patterns (learnable by a weak model) and challenging patterns (only learnable by a stronger model), as with such points, weak predictions can be used to learn challenging patterns by stronger models. We provide a practical overlap detection algorithm to find such points in datasets and leverage them to learn, among *multiple* sources of data, which to query when seeking to maximize overlap density and thereby enhance weak-to-strong generalization. We present a theoretical result showing that the generalization benefit is a function of the overlap density and a regret bound for our data selection algorithm. Empirically, we validate the mechanism and the overlap detection algorithm on a wide array of settings.

## 1 INTRODUCTION

A recurring theme in machine learning is the idea of a *less-capable entity* (e.g., a weak model or an individual with limited expertise) supervising a *stronger, more capable* one (a more powerful or higher-capacity system). The goal is to enable the stronger entity to generalize beyond the capabilities of its weaker counterpart—despite relying on its supervision. This idea undergirds classical approaches (e.g., self-training, co-training) for data-efficient learning that date back fifty years. Most recently, it drives embryonic attempts to perform *superalignment*—the process of ensuring systems with capabilities far beyond those of humans align with human values (Burns et al., 2023).

The typical flavor of works studying weak-to-strong generalization is to introduce techniques that, given a *fixed* dataset, can obtain the best performance (i.e., provide a strong model that best generalizes to an unseen test set). A vast literature has studied thousands of techniques, including entire areas, such as semi-supervised learning (Zhu & Goldberg, 2022; Ouali et al., 2020), co-training (Blum & Mitchell, 1998; Ling et al., 2009), pseudolabeling (Lee, 2013; Arazo et al., 2020), self-training (Scudder, 1965; Amini et al., 2023) student-teacher methods (Matiisen et al., 2020), and more. Much less attention, however, has been focused on what aspects of the *data* enable such techniques to succeed—and how to acquire additional data that further promotes weak-to-strong generalization.

This work focuses on this missing element. We begin by proposing a simple mechanism that captures the potential for weak-to-strong generalization. This measure, the **overlap density**, considers the potential presence of two kinds of *patterns* (i.e., sets of features or mechanisms for prediction) within each datapoint: an easy pattern—usable by weak and strong models—and a hard pattern, only accessible to the strong model. Intuitively, weak models can label points that have both patterns—the overlapping points—by taking advantage of the weak pattern (but cannot accurately label using the hard patterns). However, the strong model, using weak predictions obtained on overlapping points, can learn the harder patterns, and therefore *generalize to points **only** containing these hard patterns.*

Equipped with this intuition, we characterize the notion of overlap and provide theoretical results building on a recent theoretical framework for generalization (Lang et al., 2024). In practice, however, *overlap points are latent*, and it can be difficult to distinguish between points with *just* an easy pattern and those with overlaps. To address this challenge, we introduce an approach for identifying points with overlaps and build on it to obtain an algorithm that can, when presented with multiple sources of data, estimate which one contains the largest overlap density. This suggests a future course of action for practitioners focused on maximizing weak-to-strong generalization: ***rather than focusing on algorithms, invest in the data***—and specifically into obtaining data from sources that are likely to produce the most overlaps.

Empirically, we first validate the presence of the mechanism in a variety of real-world settings. We use the tools we proposed to identify points with overlaps. This enables us to control how much data with overlaps is included. We do so in two important application areas,

- **Weak-to-strong generalization via fine-tuning**. Here, two pretrained models are used as the weak and strong models. These have varying capacities, as in Burns et al. (2023),

- **Weak supervision.** Weak supervision (Ratner et al., 2016; 2018; Fu et al., 2020) is a framework for efficient data development. Multiple weak sources are combined via a *label model*, which serves as the weak model.

In both settings, we observe that scenarios with weak-to-strong generalization *indeed correspond to overlap density*. Next, we validate our proposed data source selection algorithm, showing enhanced data efficiency of pseudolabeled data across various datasets. We also include synthetic experiments confirming our findings and the effectiveness of our algorithm in controlled settings.

## 2    RELATED WORK

We give a brief description of related areas. Our work is complementary to many of these, as our focus is on understanding *what forms* of data promote weak-to-strong generalization—and how to obtain more of it—rather than new frameworks or training approaches. Extend related work is provided in Appendix B.

**Self-Training and Data-Efficient Learning.** Strategies that attempt to train high-quality models with less labeled data date back to the infancy of machine learning (Scudder, 1965). This idea has spawned entire fields, including semi-supervised learning (Zhu & Goldberg, 2022), weak supervision (Ratner et al., 2016; Shin et al., 2022), self-training (Amini et al., 2023; Wei et al., 2022), and more. The key distinction between such works and ours is that we are not concerned with improving performance on benchmark datasets via algorithmic improvements. Instead, we seek to understand what aspects of data result in stronger performance—and how to obtain more of the data that drives it.

**Weak-to-Strong Generalization and Superalignment.** A particularly interesting application of weak-to-strong generalization is that of *superalignment*. Superalignment, in the narrow sense, is the notion of aligning a superintelligent system to human values. More broadly, it can be thought of as aligning any large-scale system at a level of complexity beyond any individual person. As such systems may be far off into the future, researchers are currently studying *proxies*, such as smaller large language models supervising larger and more capable ones (Burns et al., 2023). Recently, several studies (Lang et al., 2024; Charikar et al., 2024; Somerstep et al., 2024) have proposed theoretical frameworks to understand weak-to-strong generalization. However, these works have yet to explore the specific data characteristics that facilitate weak-to-strong generalization. In contrast, our work provides a concrete characterization of the data that induces weak-to-strong generalization: overlap density. Building on the theoretical framework from Lang et al. (2024), we derive new theoretical results that illuminate how overlap density drives weak-to-strong generalization.

## 3    A SIMPLE DATA MECHANISM FOR WEAK-TO-STRONG GENERALIZATION

Our goal is two-fold. First, we seek to understand what properties of our data provide the possibility of weak-to-strong generalization. We introduce a simple mechanism (easy-hard overlap), formalize it, and provide a theoretical result showing that it indeed characterizes generalization.

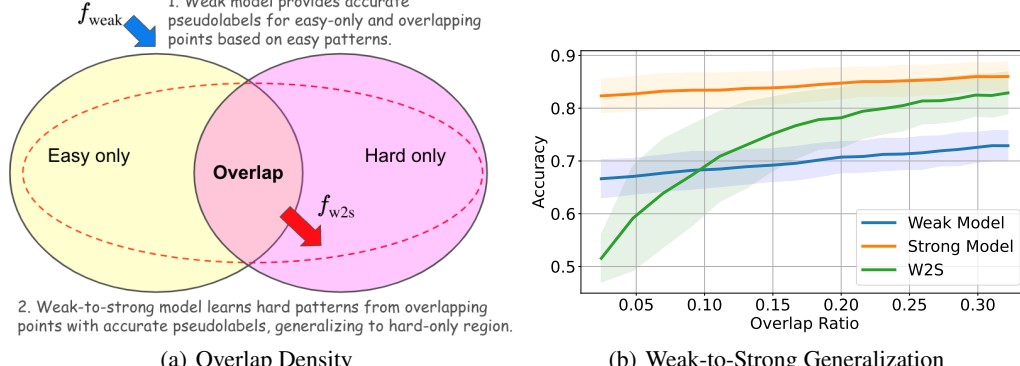

(a) Overlap Density

(b) Weak-to-Strong Generalization

Figure 1: Left: overlapping easy and hard patterns in our dataset are the key to weak-to-strong generalization. Learning from *overlapping points*, where easy features and hard features coexist, enables a weak-to-strong model $f_{w2s}$ that can generalize, while $f_{weak}$ is limited to reliably predicting points with easy patterns. Right: adding more such overlapping points has little influence on the performance of the weak model, but dramatically improves the performance of the weak-to-strong model. Adding such points—even a small percentage of the dataset—can push against the limits of the strong model.

Second, equipped with this mechanism, we wish to understand how to maximize weak-to-strong generalization. Specifically, under a data budget and with access to multiple sources of data, how can we prioritize sources that lead to greatest generalization? To address this challenge, we introduce a simple algorithm that estimates which sources have the greatest overlap density.

### 3.1 THE OVERLAP DENSITY MECHANISM

We start with an extremely simple theoretical model; afterwards, we will comment on extensions that are likely to be encountered in real-life data. Nevertheless, perhaps surprisingly, the basic mechanism often tracks weak-to-strong generalization in real settings.

**Setup.** We have access to a dataset $D_{\text{train}} = \{(x_1, y_1), \ldots, (x_n, y_n)\}$. Here, $(x, y) \sim \mathcal{D}$, $x \in \mathcal{X}$, $y \in \mathcal{Y}$, and $\mathcal{D}$ is some distribution. In addition, we have access to a dataset $D_{\text{w2s}} = \{x_{n+1}, \ldots, x_{n+m}\}$ where we do not have access to any ground-truth labels. We use two models, a weak model $f_{\text{weak}}$ and a strong model $f_{\text{w2s}}$. $f_{\text{weak}}$ is trained (or fine-tuned) on $D_{\text{train}}$ and used to output predictions $\hat{y}_j = f_{\text{weak}}(x_j)$ for points $x_j \in D_{\text{w2s}}$. $f_{\text{w2s}}$ is then trained or fine-tuned on $D_{\text{w2s}}$ with the predictions provided by $f_{\text{weak}}$. Our goal is to understand ***in what settings the strong model $f_{w2s}$ generalizes better than the weak model $f_{weak}$—despite only being trained on supervision obtained from $f_{weak}$***.

**Assumptions and Notation.** For simplicity, we will assume that $x = [x_{\text{easy}}, x_{\text{hard}}]$, where $x_{\text{easy}} \in \mathbb{R}^{d_{\text{easy}}}$ and $x_{\text{hard}} \in \mathbb{R}^{d_{\text{hard}}}$ (in practice, this is not necessary). Here, $x_{\text{easy}}$ are features producing easy patterns, learnable by the weak model, while $x_{\text{hard}}$ are hard patterns, which cannot be used by the weak model to obtain accurate predictions. In practice, feature vectors *are not a priori decomposed into such patterns*. To address this, we introduce an algorithm to estimate this identification later.

We note that any dataset $D$ can be partitioned into

- $D_{\text{overlap}}$: points containing both patterns, i.e., overlapping points,
- $D_{\text{hard only}}$: points that *only* contain the hard pattern. For convenience, in our simplified model, we take $x_{\text{easy}} = \mathbf{0}$ for such points.
- $D_{\text{easy only}}$: points that *only* contain the easy pattern. We take $x_{\text{hard}} = \mathbf{0}$ for such points.
- $D_{\text{neither}}$: points that contain neither pattern.

These four possibilities (we ignore $D_{\text{neither}}$ for simplicity) are illustrated in Fig. 1 (left). This simple categorization explains the weak-to-strong generalization phenomenon.

After training, $f_{\text{weak}}$ has learned the easy pattern, and can therefore make reliable predictions on any points in $D_{\text{overlap}} \cup D_{\text{easy only}}$, as these points contain the easy patterns. However, it will not be able to

---

**Algorithm 1 UCB-Based Data Selection for Maximizing Overlap**

---
1: **Input:** Data sources $\mathcal{D}_1, \mathcal{D}_2, \ldots, \mathcal{D}_K$, number of rounds $T \geq K$, sample size per round $n$, weak model $f_{\text{weak}}$
2: **Output:** Sampled data set $\bar{D}$ for weak-to-strong model training, Detected overlap samples $\bar{O}$
3: Try each data source once, run Overlap Detection Algorithm (Algorithm 2), Initialize $\bar{D}, \bar{O}$ with the sampled data and detected overlap data.
4: **for** $t = 1$ to $T$ **do**
5:     Compute the upper confidence bound (UCB) of overlap density in each source $s$,
        $\text{UCB}_t(s) = |\bar{O}(s)|/|\bar{D}(s)| + \sqrt{2 \log T / \bar{n}_t(s)}$, where $\bar{n}_t(s)$ is # of trials for source $s$
6:     Select the data source that maximizes UCB of overlap density.
7:     Run Algorithm 2, update $\bar{D}, \bar{O}$ with the sampled data and detected overlap data.
8: **end for**
9: **Return** $\bar{D}, \bar{O}$

---

make accurate predictions in $D_{\text{hard only}} \cup D_{\text{neither}}$. We denote the error rates as $\varepsilon_1 = \mathbb{P}(f_{\text{weak}}(x) \neq y \mid (x, y) \in D_{\text{overlap}} \cup D_{\text{easy only}})$ and $\varepsilon_2 = \mathbb{P}(f_{\text{weak}}(x) \neq y \mid (x, y) \in D_{\text{hard only}} \cup D_{\text{neither}})$. Thus, when labeling $D_{\text{w2s}}$, the predictions of $f_{\text{weak}}$ will be either reliable (in the first case), or highly unreliable (in the second case). Then, the labels for dataset $D_2$ (labeled by $f_{\text{weak}}$) are noisy with rates $\varepsilon_1, \varepsilon_2$.

> **Main assumptions**
> **(A1)**. For any $x \in \mathcal{X}$, the features of $x$ can be decomposed into easy and hard features, i.e. $x = [x_{\text{easy}}, x_{\text{hard}}]$, where $x_{\text{easy}} \in \mathbb{R}^{d_{\text{easy}}}$ and $x_{\text{hard}} \in \mathbb{R}^{d_{\text{hard}}}$.
> **(A2)**. The weak model $f_{\text{weak}}$ has no access to hard patterns, i.e., for any $x = [x_{\text{easy}}, x_{\text{hard}}]$, $f_{\text{weak}}(x) = f_{\text{weak}}(\tilde{x})$, where $\tilde{x} = [x_{\text{easy}}, \mathbf{0}]$.
> **(A3)**. We assume $\varepsilon_1 \ll \varepsilon_2$, since $f_{\text{weak}}$ cannot use hard patterns in $D_{\text{hard only}} \cup D_{\text{neither}}$.

**Generalizing Beyond the Weak Model.** Next, model $f_{\text{w2s}}$ is trained on dataset $D_{\text{w2s}}$ with its noisy labels. Since $f_{\text{w2s}}$, by assumption, has the capacity to learn the hard pattern, it can do so in the presence of noise as well. This is a well-known observation (Natarajan et al., 2013). Crucially, however, since the noise level $\varepsilon_2$ for $D_{\text{hard only}}$ is typically severe, $f_{\text{w2s}}$ can *only learn* hard patterns from points in $D_{\text{overlap}}$. As a result, the sample complexity for learning these hard patterns are given by $|D_{\text{overlap}}|$—and *not the entire dataset size $m$*.

At test time, $f_{\text{w2s}}$ has learned the easy patterns, and so will have a similar error rate as $f_{\text{weak}}$, but, unlike $f_{\text{weak}}$, will have a much smaller error rate on the hard patterns. We illustrate this idea in a synthetic scenario in Figure 1 (right). Here, we observe *three regimes*. First, when there are very few overlaps (left), the weak-to-strong model may struggle to learn the hard pattern, potentially compromising even its ability to predict easy ones. Afterwards, as the proportion of overlap points increases, the weak-to-strong model begins to dramatically improve, correctly predicting on easy points while simultaneously learning the hard patterns. Finally, the weak-to-strong model's accuracy approaches that of the strong model trained on true labels (right-most region).

## 3.2 DATA SELECTION FOR MAXIMUM OVERLAP DENSITY

A direct application of our proposed mechanism is data source selection. Specifically, we consider the following common scenario. We have access to multiple sources of data, which we call $\mathcal{D}_1, \ldots, \mathcal{D}_K$. Given a limited budget to obtain unlabeled data points from these sources, which ones should we prioritize? Clearly, to maximize weak-to-strong generalization, we should target the sources $\mathcal{D}_i$ with the largest overlap density. However, we face two challenges: **(C1)** we do not know a priori which sources have this property, and, **(C2)** even with access to data from these sources, *overlaps are latent*. That is, it is not clear how to distinguish between points in $D_{\text{overlap}}$ and points in $D_{\text{easy only}}$—as weak and strong models are both capable of accurate predictions on such points.

We propose Algorithm 1 to address these two challenges. For C1, we leverage stochastic bandit algorithms (Lattimore & Szepesvári, 2020), which balance exploration and exploitation. Here, data sources act as arms, and their average reward correspond to overlap density. Using the UCB (Upper Confidence Bound) algorithm (Auer, 2002), we explore underutilized data sources while exploiting

---

**Algorithm 2 Overlap Detection Algorithm**

---

1: **Input:** Pseudolabeled dataset $D_{\text{w2s}}$, weak model $f_{\text{weak}}$
2: **Output:** Overlap dataset, $D_{\text{overlap}}$
3: **Step 1: Separate Hard-only Points Using Confidence Scores**
4: Calculate confidence scores $\{c_i\}_{i=1}^{|D_{\text{w2s}}|}$ , where $c_i = \arg\max_j \mathbb{P}(f_{\text{weak}}(x_i) = j)$.
5: Identify hard-only points with threshold $\tau_{\text{hard}} = \text{ChangePointDetection}\Big(\{c_i\}_{i=1}^{|D_{\text{w2s}}|}\Big)$:

   • $D_{\text{hard only}} = \{x_i \in D_{w2s} \mid c_i \leq \tau_{\text{hard}}\}$          # Low confidence
   • $D_{\text{non-hard only}} = D_{w2s} \setminus D_{\text{hard only}}$          # High confidence
6: **Step 2: Identify Overlapping Points from $D_{\text{non-hard only}}$**
7: Calculate overlap scores $s_i = \max\{|x_i^T x_{\text{hard}}| | x_{\text{hard}} \in D_{\text{hard only}}\}$ for each $x_i \in D_{\text{non-hard only}}$
8: Identify overlap points with threshold $\tau_{\text{overlap}} = \text{ChangePointDetection}\Big(\{s_i\}_{i=1}^{|D_{\text{non-hard only}}|}\Big)$:

   • $D_{\text{overlap}} = \{x_i \in D_{\text{non-hard only}} | s_i \geq \tau_{\text{overlap}}\}$          # High overlap scores
9: **return** $D_{\text{overlap}}$

---

those with high overlap density. Each data source $\mathcal{D}_s$ has an overlap density $o_s$, influenced by noise from the sampling process and overlap detection. Initially, we sample each source once. In subsequent iterations, we choose the source with the highest UCB. This is computed as the sum of the estimated overlap density and a confidence radius that promotes exploration (Algorithm A1).

To estimate the overlap densities, we must address C2. We propose an overlap detection algorithm (Algorithm 2) based on two insights from our data model in Section 3.1, i.e. $x = [x_{\text{easy}}, x_{\text{hard}}]$.

1. Weak models are less confident on hard-only points because they lacks access to hard features.

2. Overlap points are more closely aligned with hard-only points than easy-only points are.

We provide theoretical support for these in Section 4.2. Based on these, we use confidence scores to separate hard-only data points first, and then use the absolute values of inner products as overlap scores to distinguish overlap points from easy-only points. In Algorithm 2, we first identify hard-only points by thresholding weak model confidence scores, split the dataset, and detect overlap points using inner products to distinguish them from easy-only points. We determine thresholds using a change point detection technique (Sen & Srivastava, 1975). The intuitions underlying our algorithms are empirically validated in the experiments in Section 5.1, where our overlap detection algorithm effectively isolates overlap points, leading to improved generalization.

## 4 THEORETICAL ANALYSIS

We introduce a theoretical result showing that weak-to-strong generalization is governed by the overlap density. Afterwards, we provide and interpret theoretical guarantees for our overlap detection and data source selection algorithms.

### 4.1 WEAK-TO-STRONG GENERALIZATION VIA OVERLAP EXPANSION

We build off of the framework in Lang et al. (2024), where generalization is governed by an *expansion* property. Specifically, we show that the weak-to-strong model can correct the weak model's pseudolabels on hard data points $D_{\text{hard only}}$, since the pseudolabels produced by the weak model are (relatively) accurate on overlapping points and the strong model can learn hard patterns that address hard data points. We first introduce the relevant definitions and outline our setup.

**Definition 1** (Expansion). *(Lang et al., 2024) Fix sets $A, B \subset \mathcal{X}$ and a neighborhood function $\mathcal{N}$. We say the distribution $\mathbb{P}_{\mathbf{x}}$ satisfies $(c,q)$-expansion on $(A, B)$ if for all sets $U \subset B$ with $\mathbb{P}(U|B) > q$, $\mathbb{P}(\mathcal{N}(U)|A) > c\mathbb{P}(U|B)$.*

**Definition 2** ($\eta$-robust). *(Lang et al., 2024) For a classifier $f$ and a point $x$, define $r(f, x) = \mathbb{P}(f(\mathbf{x}') \neq f(x)|\mathbf{x}' \in \mathcal{N}(x))$ as the probability of label disagreement between $x$ and its neighbor $\mathbf{x}'$. A classifier $f$ is $\eta$-robust at $x$ if $r(f, x) \leq \eta$. The set of $\eta$-robust points for $f$ is $R_\eta(f) = \{x : r(f, x) \leq \eta\}$.*

**Problem Setup.** We use the setup described in Section 3.1. Additionally, let $S_i$ represent the dataset whose labels are class $i$, $S_i^{\text{good}}$ be the correctly pseudolabeled subset of $S_i$, and $S_i^{\text{bad}}$ the incorrectly pseudolabeled subset of $S_i$. With a little abuse of notation, we denote the true labeling function as $y$. Our goal is to show how overlap density translates into the strong model's generalization on points in $D_{\text{hard only}}$ via the expansion property. Our key assumption is that the data distribution and the weak-to-strong model behavior on $S_i^{\text{good}} \cap D_{\text{overlap}}$ expands through the neighborhood structure to $S_i^{\text{bad}} \cap D_{\text{hard only}}$, where the weak model struggles with hard patterns, relying on robustness. This assumption captures the intuition that the strong model can learn patterns usable for predicting the hard points from the overlap points. We state a simplified version of the theorem first; the full version is in the Appendix D.1.

**Theorem 4.1.** *Suppose $\mathbb{P}$ satisfies $(c, q)$ expansion on $(S_i^{bad} \cap D_{hard\,only}, S_i^{good} \cap D_{overlap})$ for some $c > 0$. Consider an arbitrary $\eta$-robust classifier $f_{w2s}$ such that $\mathbb{P}(f_{w2s}(\mathbf{x}) \neq f_{weak}(\mathbf{x})$ at $\mathbf{x}|S_i \cap D_{overlap}) \leq 1 - q - \varepsilon_1$. Then, the classifier $f_{w2s}$, $f_{w2s}$ satisfies the following error bound:*

$$err(f_{w2s}, y|S_i \cap D_{hard\,only}) \leq err(f_{w2s}, f_{weak}|S_i \cap D_{hard\,only}) + \varepsilon_2$$
$$- 2c\varepsilon_2(1 - err(f_{w2s}, f_{weak}|S_i^{good} \cap D_{overlap}) - \mathbb{P}(R_\eta(f_{w2s})^c|S_i^{good} \cap D_{overlap}))$$

The full statement and proof are provided in Appendix D.1, and additional coverage expansion result is provided in Appendix D.2. We note that the bound is highly dependent on the neighborhood function $\mathcal{N}$ which determines the parameters $c$ and $q$. To understand the role of $q$, suppose that for any fixed $q \in (0, 1)$, $c$ is optimal (i.e. any smaller value of $c$ fails the expansion criterion). Then, increasing $q$ will cause $c$ to increase as well, but we have a constraint $q \leq 1 - \mathbb{P}(f_{\text{w2s}}(\mathbf{x}) \neq f_{\text{weak}}(\mathbf{x})$ at $\mathbf{x}|S_i \cap D_{\text{overlap}}) - \varepsilon_1$ from $\eta$-robust condition, which subsequently constrains c as well.

This bound demonstrates that weak-to-strong generalization is achievable *as long as the overlap density expands to a sufficient extent* (i.e., large $c$), the error rate in estimating the correct overlap density is low $\left(\text{i.e., small err}(f_{\text{w2s}}, f_{\text{weak}}|S_i^{\text{good}} \cap D_{\text{overlap}})\right)$, and $f_{w2s}$ is adversarially robust $\left(\text{i.e. small } \mathbb{P}(R_\eta(f_{\text{w2s}})^c|S_i^{good} \cap D_{\text{overlap}})\right)$. Specifically, when $f_{\text{w2s}}$ exactly replicates $f_{\text{weak}}$, we have $\text{err}(f_{\text{w2s}}, y|S_i \cap D_{\text{hard only}}) = \varepsilon_2$. We aim for a tighter bound than this. Pseudolabel correction is provably achieved when the right-hand side is less than $\varepsilon_2$, and the improvement of the bound over $\varepsilon_2$ can be quantified as

$$\rho = 2c\varepsilon_2 \left(1 - \text{err}(f_{\text{w2s}}, f_{\text{weak}}|S_i^{good} \cap D_{\text{overlap}}) - \mathbb{P}(R_\eta(f_{\text{w2s}})^c|S_i^{good} \cap D_{\text{overlap}})\right)$$
$$- \text{err}(f_{\text{w2s}}, f_{\text{weak}}|S_i \cap D_{\text{hard only}}).$$

This result largely follows from the framework in Lang et al. (2024); the upshot is that the overlap density mechanism is consistent with existing frameworks for weak-to-strong generalization. However, as we shall soon see, it offers a critical advantage: it permits us to operate with a data-centric perspective that enables users to *improve weak-to-strong generalization.*

## 4.2 THEORETICAL GUARANTEES FOR OVERLAP DETECTION AND DATA SELECTION

Equipped with the previous result, we provide a theoretical guarantee of our overlap detection algorithm under a Gaussian mixture assumption. We derive a regret bound of our UCB-based data selection algorithm for overlap density maximization.

**Overlap Detection.** We provide a theoretical guarantee for the overlap score under the assumptions described in Section 3.1, i.e. $x = [x_{\text{easy}}, x_{\text{hard}}]$, where $x_{\text{easy}} \in \mathbb{R}^{d_{\text{easy}}}$ and $x_{\text{hard}} \in \mathbb{R}^{d_{\text{hard}}}$. Let $\tilde{x} = g(x) = [x_{\text{easy}}, \mathbf{0}]$ represent the input vector for the weak model, where hard features from $x$ are zeroed out. More detailed setup specific to this section is described in Appendix D.3. For $x \in D_{\text{hard only}}$, $\tilde{x} = [\mathbf{0}, \mathbf{0}]$, so the weak model prediction probability is $f_{\text{weak}}(x) = \sigma(\theta^\top \tilde{x}) = \sigma(0) = 0.5$, which corresponds to the minimum confidence score. This ensures the perfect accuracy of detecting hard-only points using Algorithm 2 with $\tau_{\text{hard}} = 0.5$. Next, we aim to separate overlap points from easy-only points. Under the Gaussian mixture assumption in D.3, we have $\mathbf{x}_{\text{easy only}} \sim \mathbf{N}(\mu_{\text{easy}}, cI)$, and $\mathbf{x}_{\text{overlap}} \sim \mathbf{N}(\mu_{\text{overlap}}, cI)$, where $\mu_{\text{overlap}} = [\tilde{\mu}_{\text{easy}}, \tilde{\mu}_{\text{hard}}]^\top$, $\mu_{\text{easy}} = [\tilde{\mu}_{\text{easy}}, 0]^\top$. To show the effectiveness of overlap separation from easy-only points in Algorithm 2, we demonstrate that $\mathbf{x}_{\text{overlap}}^\top \mathbf{x}_{\text{hard only}}$ and $\mathbf{x}_{\text{easy only}}^\top \mathbf{x}_{\text{hard only}}$ exhibit a distributionally distinguishable gap.

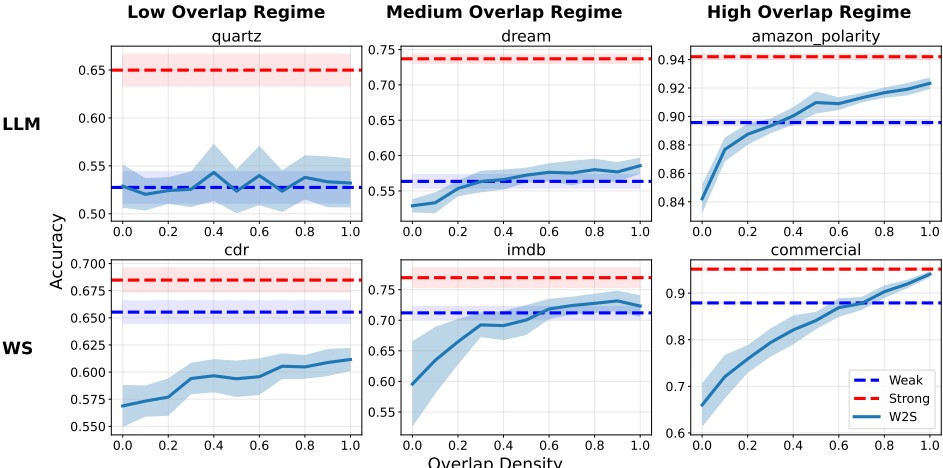

Figure 2: Overlap density versus performance in weak-to-strong generalization with LLMs. Red lines show strong ceiling model accuracies, blue dashed lines represent weak model test accuracies, and W2S lines represent the accuracies of strong models trained on pseudolabeled data with a controlled proportion of overlap density. In general, ***the strong model's improvement over the weak model tracks the overlap proportion, suggesting that the overlap density is indeed an important mechanism for generalization***. We can observe three different regimes of weak-to-strong generalization in our experiments: a **low overlap regime**, where the overlap density is insufficient for effective weak-to-strong generalization (here, few points contain overlaps, so choosing to rely on a large overlap proportion translates to a small train set), a **medium overlap regime**, where the overlap density improves generalization but still yields performance close to that of the weak model, and a **high-overlap regime**, where the strong model's performance approaches that of the true strong model due to sufficient overlap points.

**Theorem 4.2.** *Given the above setup,* $\mathbb{E}[\mathbf{x}_{overlap}^\top \mathbf{x}_{hard\,only}] - \mathbb{E}[\mathbf{x}_{easy\,only}^\top \mathbf{x}_{hard\,only}] = \|\mu_{hard}\|_2^2$. *Furthermore, we have*

$$\mathbb{P}\left(\mathbf{x}_{overlap}^\top \mathbf{x}_{hard\,only} \le \mathbf{x}_{easy\,only}^\top \mathbf{x}_{hard\,only}\right) \le \exp\left(-\min\left(\frac{3\|\mu_{hard}\|_2^4}{16dc^2 + 18c\|\mu_{hard}\|_2^2}, \frac{\|\mu_{hard}\|_2^2}{8c}\right)\right).$$

This result shows the average gap $\|\mu_{hard}\|_2^2$ between $\mathbf{x}_{overlap}^\top \mathbf{x}_{hard\,only}$ and $\mathbf{x}_{easy\,only}^\top \mathbf{x}_{hard\,only}$ and the bound on the probability that the overlap score of an easy-only point exceeds that of an overlap point. The error bound result implies that the accuracy of the overlap detection algorithm deteriorates as the noise level $c$ and dimension $d$ increase. The proof is provided in Appendix D.3.

**Data Selection.** We provide a regret bound for our data selection algorithm (Algorithm 1), which quantifies the gap in overlap density between selecting the optimal data source in every round and using our algorithm, which balances exploration and exploitation through the UCB algorithm. Let $o(s) = \mathbb{P}(\mathcal{D}_{overlap}|\mathcal{D}_s)$ be the population overlap density of source $s$, $\bar{o}_t(s) = |\bar{O}_t(s)|/|\bar{D}_t(s)|$ be the empirical overlap density of source $s$ at round $t$, and $o^* = \max_s o(s)$ be the optimal overlap density. The following theorem establishes an upper bound on the expected average regret at round $t$, $\mathbb{E}[o^* - \bar{o}_t]$.

**Theorem 4.3.** $\mathbb{E}[o^* - \bar{o}_t] \le O\left(\sqrt{K\log T/t}\right)$, *where $K$ is the number of data sources, $T$ is the total number of rounds.*

For the details, refer to Appendix D.4. This result shows that the gap between the optimal and the empirical overlap ratio obtained with Algorithm 1 decreases at a rate of $O\left(\sqrt{\log T/T}\right)$ in $T$. This implies that $\mathbb{E}[\bar{o}_T] \to o^*(T)$ as $T \to \infty$.

## 5 EXPERIMENTS

We first validate the role of the data overlap mechanism in weak-to-strong generalization, examining two cases: large language models following the setup of Burns et al. (2023) and the weak supervision setting, where the weak model is a label model, often a probabilistic graphical model. Next, we evaluate the effectiveness of our UCB-based data selection strategy from Algorithm 1. Afterwards, we confirm our theoretical claims in a controlled setting, showing that the performance gains from overlap density primarily benefit hard-only data points, and that our data selection algorithm maximizes overlap density, improving weak-to-strong generalization. Our code is available at https://github.com/SprocketLab/datacentric_w2s.

### 5.1 WEAK-TO-STRONG GENERALIZATION VIA OVERLAP DENSITY MECHANISM

We follow the approach in Burns et al. (2023), where the goal is to use large language models as proxies for weak agents supervising superintelligent agents. Our hypothesis is that the overlap density mechanism elicits weak-to-strong generalization in this setting. We anticipate that a higher overlap density generally enhances the performance of weak-to-strong models. Additionally, we hypothesize the existence of three regimes of weak-to-strong generalization from datasets can be observed depending on the amount of overlap data points in the dataset and the noise level of overlap detection.

- **Low overlap regime**: Insufficient overlap points or overly noisy detection hinder weak-to-strong generalization, leading to performance worse than $f_{\text{weak}}$.
- **Medium overlap regime**: Adequate overlap points and moderate noise levels enable weak-to-strong generalization, resulting in performance comparable to, or slightly better than, $f_{\text{weak}}$.
- **High overlap regime**: Sufficient overlap points with minimal noise in overlap detection induce strong weak-to-strong generalization, with performance approaching $f_{\text{strong}}$.

**Setup and Procedure.** We split the original training data into two subsets, $D_{\text{train}}$ and $D_{\text{w2s}}$. The weak models are trained on $D_{\text{train}}$ and then generate weak labels for $D_{\text{w2s}}$. The weak-to-strong models are subsequently trained on $D_{\text{w2s}}$ using these weak labels. Using the overlap detection algorithm (Algorithm 2), we identified subsets $\hat{D}_{\text{overlap}}$ and $\hat{D}_{\text{nonoverlap}}$, and sampled $n_{\text{controlled}}$ data points to control overlap density between 0% and 100%, creating the dataset $D_{\text{controlled},\alpha}$, where $\alpha$ denotes the overlap ratio. The weak-to-strong (W2S) models were then trained on $D_{\text{controlled},\alpha}$. ***Crucially, if the total quantity of overlap points is small (i.e., because the overlap density is small), building a dataset whose ratio is high translates into fewer overall points for training.*** Details on the distribution of detected easy-only, hard-only, and overlap points can be found in Appendix E.

For the language model experiments, we followed the setup described in EleutherAI (2021), which replicates Burns et al. (2023). We used the Qwen1.5 0.5B model as the weak model and the Llama3 8B model as the strong model. We used linear probing based on the observation in Appendix D.2 of Burns et al. (2023) that linear probing results often align with those from full fine-tuning. We used 19 datasets from EleutherAI (2021). For the weak supervision setting, we used 9 datasets from the WRENCH weak supervision benchmark (Zhang et al., 2021). We used Snorkel (Ratner et al., 2018) as the label model (weak model), and a 4-layer MLP was used as the strong model.

**Results.** Figure 2 presents the results of this experiment. As expected, we observe that the strong model performance improves as the overlap proportion increases, *providing evidence for the overlap density mechanism's role* in weak-to-strong generalization. Also, we were able to observe three regimes of weak-to-strong generalization by our overlap detection method. We showcased each case in LLM and weak supervision settings, respectively. Full experimental results are provided in Appendix F.1. Additionally, the ablation study on model architecture in the weak supervision setting is presented in Appendix F.5, and the transferability study in Appendix F.6.

### 5.2 DATA SOURCE SELECTION VIA OVERLAP DETECTION

Next, we validate our data selection procedure instantiated in Algorithm 1. We hypothesize that the UCB-based overlap maximization strategy leads to better weak-to-strong generalization by identifying the optimal data source given multiple sources with varying overlap densities.

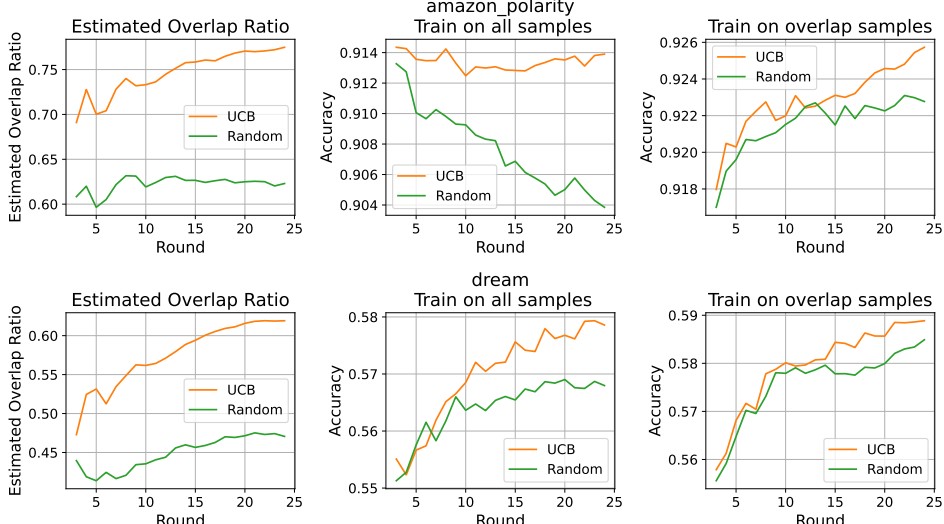

Figure 3: Data selection results with Algorithm 1 for Amazon Polarity and DREAM datasets. We report the average of 20 repeated experiments with different seeds. We observe that the data source selection procedure, based on overlap density estimation, can produce enhancements over random sampling across data sources.

**Setup and Procedure.** We used a similar setup and datasets as in the large language model experiments. Overlap density in the training sets was identified using our proposed method (Algorithm 2), and the weak-to-strong training dataset was split into $D_1$ and $D_2$ with overlap densities of 0.9 and 0.1, respectively. With these two data sources, we ran Algorithm 1 and compared its performance to random sampling. The number of rounds was set to $T = 25$.

**Results.** The results are presented in Figure 3. We observe that the UCB-based overlap maximization algorithm can lead to better weak-to-strong generalization. We note that we do not *always* expect to obtain results such as those in Figure 3. Indeed, if there simply are very few overlap points—or if the procedure for identifying them is very noisy — we will not observe these types of results—or any type of weak-to-strong generalization. Full experimental results are provided in Appendix F.2.

### 5.3 SYNTHETIC EXPERIMENTS

We verify our overlap density mechanism and data selection algorithm in fully controllable settings.

#### 5.3.1 OVERLAP DENSITY MECHANISM

We validate the claim that overlap density enhances the performance of a weak-to-strong model on hard data points, while the weak model exhibits random accuracy on those same points.

**Setup and Procedure.** We simulate weak-to-strong generalization using a simple mixture of Gaussians setup with logistic regression models, as described in Section 3. In this setup, the hard features are intentionally blocked (set to **0**) for the weak models to mimic the common scenario where weak models lack access to these features. Full details are provided in Appendix E. The weak model $f_{\text{weak}}$ is trained on a dataset $D_{\text{train}}$ and generates pseudolabels for $D_{\text{w2s}}$. The weak-to-strong model $f_{\text{w2s}}$ is then trained on $D_{\text{w2s}}$ using these pseudolabels and evaluated on $D_{\text{test}}$. We set $n_{\text{easy}} = 100$ and $n_{\text{hard}} = 100$, incrementing $n_{\text{overlap}}$ by 5 in each iteration. The performance of the weak-to-strong model is assessed on easy-only, hard-only, and overlap data points in the test set, respectively.

**Results.** Figure 4 illustrates how accuracy varies with the overlap ratio across easy-only, hard-only, and overlap data points. As expected, the most substantial performance improvement over the weak pseudolabeler occurs on the hard-only data points. On the easy-only data, the weak-to-strong model initially underperforms compared to the weak model due to label noise. However, as the overlap density increases, the weak-to-strong model's performance approaches that of the weak model. On

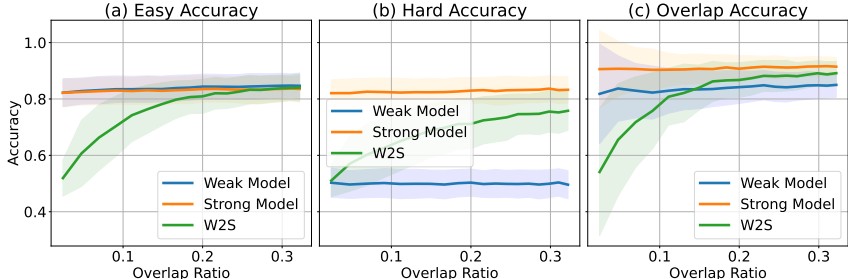

Figure 4: Accuracy in each data region in synthetic experiments. As expected, the performance gain mainly comes from hard data points as the overlap density increases.

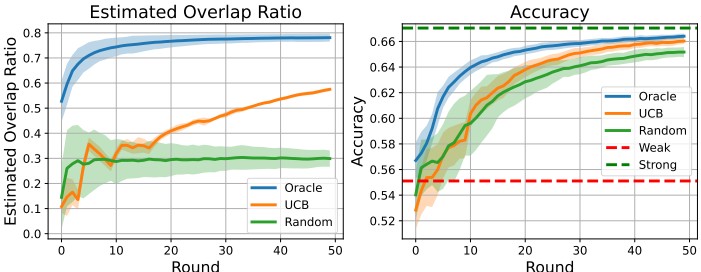

Figure 5: Synthetic data selection experiment. Our algorithm demonstrates better data efficiency than random sampling by consistently identifying the data source with the highest overlap density.

overlap data points, the weak-to-strong model also starts off performing worse due to similar label noise, but as it learns hard patterns, it eventually outperforms the weak model.

### 5.3.2 DATA SOURCE SELECTION

We validate the claim that the algorithm effectively identifies data sources with higher overlap density and progressively maximizes it with each round, leading to improved weak-to-strong generalization.

**Setup.** We set $K = 5$ data sources, each characterized by different overlap densities: [0.1, 0.15, 0.2, 0.05, 0.8]. For the nonoverlap distribution, we assumed that the half of the nonoverlap distribution is easy-only, and the rest half is hard-only. We compare our data source selection algorithm against random sampling and an oracle setting, where data are always sampled from the data source with the optmal overlap density. In each round, 100 data points were sampled from the selected data source, with the total number of rounds set to $T = 50$.

**Results.** Figure 5 presents the experimental results. As the rounds progress, our algorithm increasingly identifies the optimal data source, achieving better weak-to-strong generalization compared to random sampling. Additional experimental results on easy-only and hard-only training are provided in Appendix F.3, along with a sensitivity analysis for overlap detection noise in Appendix F.4.

## 6 CONCLUSION

We studied a data-centric mechanism that explains weak-to-strong generalization. This mechanism is centered on easy-hard overlaps: points where weak models leverage easy patterns for accurate labeling, while strong models learn hard patterns to generalize to hard-only points. We studied this idea conceptually, theoretically, and empirically; the latter in multiple popular settings. Finally, we introduced algorithms for identifying overlapping points and determining, given a limited data budget, which data sources should be queried to maximize weak-to-strong generalization.

Our study was limited to a simple version of what is likely to be a more complex mechanism in many realistic settings. We are interested in extending this work in several directions. These include allowing more complex patterns (multiple levels of overlapping difficulties), further theoretical results, and studying additional variations for the overlap identification procedure.

ACKNOWLEDGMENTS

We are grateful for the support of the National Science Foundation (NSF) (CCF2106707), the Defense Advanced Research Projects Agency (DARPA Young Faculty Award), and the Wisconsin Alumni Research Foundation (WARF).

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

APPENDIX

The appendix contains additional details, proofs, and experimental results. The glossary (Appendix A) contains a convenient reminder of our terminology. Appendix B provides more related works and discussion about the relationship between our work and related papers. In Appendix C, we describe the details of our algorithms and discuss their implementations. Appendix D provides the proofs of theorems that appeared in Section 4. Finally, we provide additional details and analysis of the experiments in Appendix E and present further experimental results in Appendix F.

## A  GLOSSARY

The glossary is given in Table A1.

| Symbol | Definition |
| --- | --- |
| $n$ | Total number of samples |
| $\mathcal{X}$ | Feature space |
| $\mathcal{Y}$ | Label space |
| $y(\cdot)$ | Underlying labeling function |
| $\mathcal{D}$ | Data distribution |
| $D_{\text{easy only}}$ | Set of data points that only contain the easy patterns |
| $D_{\text{hard only}}$ | Set of data points that only contain the hard patterns |
| $D_{\text{overlap}}$ | Set of data points containing both easy and hard patterns |
| $D_{\text{train}}$ | Labeled training set for weak model |
| $D_{\text{w2s}}$ | Unlabeled or pseudolabeled training set for weak-to-strong models |
| $D_{\text{controlled},\alpha}$ | Sampled dataset from $D_{\text{w2s}}$ with the controlled overlap ratio as $\alpha$ |
| $f_{\text{weak}}$ | Weak model |
| $f_{\text{strong}}$ | Strong model fine-tuned with true labels |
| $f_{\text{w2s}}$ | Weak-to-strong model (Strong model fine-tuned with pseudolabels generated by weak model) |
| $\varepsilon_1$ | Error rate of weak model in easy-only and overlap, $\mathbb{P}(f_{\text{weak}}(x) \neq y \mid (x,y) \in D_{\text{overlap}} \cup D_{\text{easy only}})$ |
| $\varepsilon_2$ | Error rate of weak model in hard-only points, $\mathbb{P}(f_{\text{weak}}(x) \neq y \mid (x,y) \in D_{\text{hard only}} \cup D_{\text{neither}})$ |
| $S_i$ | Covered data points of class $i$ |
| $S_i^{\text{good}}$ | Correctly pseudolabeled data points of class $i$ |
| $S_i^{\text{bad}}$ | Incorrectly pseudolabeled data points of class $i$ |
| $T_i$ | Uncovered data points of class $i$ |
| $o(s)$ | Overlap density for a specific data source $s$ |
| $o^*$ | Best overlap density of all the data sources |
| $\bar{o}_t(s)$ | Empirical overlap density of data source $s$ at round $t$ |
| $T$ | Number of data selection rounds |
| $\mathcal{D}_i$ | $i$-th data source |
| $\bar{D}$ | Sampled data from data sources |
| $\bar{O}$ | Detected overlap points in $\bar{D}$ |
| $\bar{D}_t(s)$ | Sampled data from data source $s$ up to round $t$ |
| $\bar{O}_t(s)$ | Detected overlap points sampled from data source $s$ up to round $t$ |

Table A1: Glossary

## B  EXTENDED RELATED WORK

**Theory of Weak-To-Strong Generalization.**  Several theoretical approaches have been proposed to explain weak-to-strong generalization, reflecting growing interest in this area of research. Somerstep et al. (2024) frame weak-to-strong generalization as a transfer learning problem, where latent knowledge from weak models is transferred to strong models. They propose label refinement to address the limitations of naive fine-tuning on pseudolabels, i.e. naive fine-tuning often leads to the

strong model replicating the errors of the weak models. Charikar et al. (2024) present a theoretical framework rooted in convex analysis to explain weak-to-strong generalization. They suggest that the disagreements between strong models and weak models' pseudolabels predicts weak-to-strong generalization performance. This aligns with our theory, where a strong model trained on overlap density corrects pseudolabels on hard data points. Finally, Lang et al. (2024) introduce a framework that explains pseudolabel correction and coverage expansion based on the expansion property and model robustness. We specifically characterize where such expansions occur and provide both theoretical and empirical validation of how they lead to weak-to-strong generalization.

**Evaluating the Difficulty of Data Points.** The idea that certain points are more difficult than others has a long history. There are several ways to define hardness in supervised learning alone. These include closeness to the decision boundary, as in active learning (Dasgupta et al., 2005; Hanneke, 2014), low-confidence points, as in selective classification (El-Yaniv & Wiener, 2010; Geifman & El-Yaniv, 2017), points (or sets of points) with high values of the loss function, and heuristic rules applied to the data (Sun et al., 2024). Agarwal et al. (2022) assess sample difficulty by examining the variance of gradients during training. Baldock et al. (2021) employ deep neural networks, operating under the intuition that difficult samples are more likely to be predicted in higher layers when using linear probing at each layer. Seedat et al. (2024) provide a comprehensive survey and taxonomy of data hardness characterization, along with a benchmark for comparing different methods. However, none of these studies address overlapping data points, which contain both easy and hard patterns and could be crucial for understanding weak-to-strong generalization.

**Data Valuation.** A large number of works have studied ways to understand the impact of each point in a training dataset on the performance of the resulting model. These include approaches via influence functions Koh & Liang (2017), Shapley values Ghorbani & Zou (2019), and other methods Karr et al. (2006); Yoon et al. (2020). The main difference between these works and ours is that we are less concerned with any single point and model training in general, but with an *overall data mechanism* that is relevant to the weak-to-strong generalization setting.

**Data Curation and Selection** Selecting subsets of data to achieve better performance with less data has been extensively studied, particularly in the context of foundation model training. Xie et al. (2023) introduce Data Selection with Importance Resampling (DSIR), which selects pretraining data for language models by estimating importance weights in a reduced feature space, aligning the selected data distribution with a desired target distribution. Similarly, Ankner et al. (2024) use the perplexity of smaller models to curate pretraining datasets, while Wettig et al. (2024) employ quality qualifiers to assign scores and steer data curation process. Xia et al. (2024) use influence functions to curate instruction-tuning datasets. Finally, Li et al. (2024) present DataComp for Language Models (DCLM), a benchmark designed to evaluate dataset curation strategies within a general data curation pipeline for pretraining large language models. Our overlap detection process aligns with the broader theme of data curation but is specifically tailored to the context of weak-to-strong generalization.

In a more general context, Campi & Garatti (2023) introduce a theoretical framework for compression functions in learning, laying the foundation for data selection methods. Building on this framework, Paccagnan et al. (2024) propose Pick2Learn (P2L) meta-algorithmic framework, which iteratively selects data subsets until a specified appropriateness criterion is satisfied, producing a compressed dataset that preserves most of the relevant information. While the concept of overlap density bears a conceptual resemblance to the output of the P2L algorithm—both aim to identify data subsets that generalize effectively, overlap density explicitly addresses the noise inherent in pseudolabels generated by weak supervisors, making it particularly suited for weak-to-strong generalization. Nonetheless, P2L algorithm could potentially be adapted for overlap detection, provided an appropriate criterion for overlap identification is defined within its framework.

## C  ALGORITHM DETAILS

We provide the detailed version of Algorithm 1 in Algorithm A1, and discuss details of Algorithm 2.

**Threshold selection.** For the thresholds in Algorithm 2, we used a change point detection algorithm, specifically the binary segmentation method (Sen & Srivastava, 1975), applied to the sorted confidence

---

**Algorithm A1 UCB-Based Data Selection for Maximizing Overlap (Detailed)**

---

1: **Input:** Data sources $\mathcal{D}_1, \mathcal{D}_2, \ldots, \mathcal{D}_K$, the number of rounds: $T \geq K$, sample size per round: $n$, weak model: $f_{\text{weak}}$
2: **Output:** Sampled data set $\bar{D}$ for weak-to-strong model training, Detected overlap samples $\bar{O}$
3: **Initialization:** $\bar{D} = \emptyset, \bar{O} = \emptyset$
4: # Try each data source once
5: **for** $t = 1$ to $K$ **do**
6:     $k \leftarrow t$, Sample $n$ points $S_t(k) = \{x_1, \ldots, x_n\}$ from $D_k$
7:     Do pseudolabeling of samples using the weak model: $S_t(k) \leftarrow \{(x, f_{\text{weak}}(x)) \mid x \in S_t(k)\}$
8:     Initialize $\bar{D}_t(k) \leftarrow S_t(k)$, Update $\bar{D} \leftarrow \bar{D} \cup S_t(k)$
9:     Detect overlap points $O_t(k)$ in $\bar{S}_t(k)$ using Algorithm 2, initialize $\bar{O}_t(k) \leftarrow O_t(k)$, $\bar{O} \leftarrow O_t(k)$.
10:     Initialize choice count: $\bar{n}_t(k) \leftarrow 1$
11: **end for**
12:
13: # Choose data source in each round based on UCB (Upper Confidence Bound)
14: **for** $t = K + 1$ to $T$ **do**
15:     **For each** $k = 1$ to $K$ **do**
16:         Compute UCB score: $\text{UCB}_t(k) \leftarrow \dfrac{|\bar{O}_{t-1}(k)|}{|\bar{D}_{t-1}(k)|} + \sqrt{\dfrac{2 \log T}{\bar{n}_t(k)}}$
17:     Select data source: $k^* \leftarrow \arg\max_k \text{UCB}_t(k)$
18:     Sample $n$ points $S_t(k) = \{x_1, \ldots, x_n\}$ from $D_k$
19:     Label new samples using $f_{\text{weak}}$ and do pseudolabeling with $f_{\text{weak}}$
20:     Detect overlap points $O_t(k)$ in $\bar{S}_t(k)$ using Algorithm 2
21:     $\bar{D}_t(k^*) \leftarrow \bar{D}_{t-1}(k^*) \cup S$, $\bar{D}_t(k) \leftarrow \bar{D}_{t-1}(k)$ for $k \neq k^*$, $\bar{D} \leftarrow \bar{D} \cup S$
22:     $\bar{O}_t(k^*) \leftarrow \bar{O}_{t-1}(k^*) \cup O$, $\bar{O}_t(k) \leftarrow \bar{O}_{t-1}(k)$ for $k \neq k^*$, $\bar{O} \leftarrow \bar{O} \cup O$
23:     Increment choice count: $\bar{n}_t(k^*) \leftarrow \bar{n}_{t-1}(k^*) + 1$, $\bar{n}_t(k) \leftarrow \bar{n}_{t-1}(k)$ for $k \neq k^*$
24: **end for**
25: **Return** $\bar{D}, \bar{O}$

---

and overlap scores. The implementation used the ruptures Python package (Truong et al., 2020). Change point detection methods identify points where statistical properties of a data sequence shift. In our approach, we assume that the distributions of confidence scores and overlap scores vary across hard-only, easy-only, and overlap regions, allowing the change point detection method to identify these differences. Alternatively, segmentation methods like K-means clustering could also be used for this purpose.

**Overlap scoring.** While we provide a theory for inner product scores and taking maximum as our algorithm to identify overlap density, we used the absolute value of the cosine similarity in large language model experiments and weak supervision experiments after testing various statistics like mean, median, 75th percentile, and Euclidean distance instead of the inner product. The core intuition is that overlap points are closer in distribution to hard-only points than to easy-only points. We anticipate other measures could capture this intuition, with influence functions (Koh & Liang, 2017) being a promising alternative for future work. Additionally, when using neural networks, we computed overlap points from the last layer activations rather than the inputs, as this provided clearer signals in practice.

# D  THEORY DETAILS

## D.1  PROOF OF THEOREM 4.1

We provide a theoretical result building off Lang et al. (2024). The main idea is that the strong model can generalize better by learning from overlap data points, which leads to the expansion property and pseudolabel correction phenomenon in Lang et al. (2024). We begin by adopting the definitions from Lang et al. (2024).

**Notations.** Let $\mathbf{x}$ denote a random variable with distribution $\mathcal{D}$, and $x$ represent its realizations. We assume the existence of a ground-truth labeler $y : \mathcal{X} \to \mathcal{Y} = 1, \ldots, k$ and a weak model (pseudolabeler) $f_{\text{weak}} : \mathcal{X} \to \mathcal{Y} \cup \{\varnothing\}$, which assigns to each $x$ either a label from $\mathcal{Y}$ or the abstention symbol $\varnothing$.

Define $S = \{x \mid f_{\text{weak}}(x) \neq \varnothing\}$ as the *covered* subset of $\mathcal{X}$ (the set with pseudolabels), and $T = \{x \mid f_{\text{weak}}(x) = \varnothing\} = \mathcal{X} \setminus S$ as the uncovered subset. Training occurs on the pseudolabeled *source* subset $S$, and evaluation spans both $S$ and the uncovered *target* $T$. Let $\{\mathcal{X}_i\}$ be a partition of $\mathcal{X}$ where the ground-truth label is constant in each $\mathcal{X}_i$ (e.g., $\mathcal{X}_i = \{x \mid y(x) = i\}$). We use this partition for convenience, but our results generalize to other partitions. $S$ and $T$ are partitioned into $S_i = S \cap \mathcal{X}_i$ and $T_i = T \cap \mathcal{X}_i$, respectively. Finally, each $S_i$ is divided into correctly pseudolabeled examples $S_i^{\text{good}} = \{x \in S_i \mid f_{\text{weak}}(x) = y(x)\}$ and incorrectly pseudolabeled examples $S_i^{\text{bad}} = S_i \setminus S_i^{\text{good}}$.

**Definitions.** For two classifiers $f, g : \mathcal{X} \to \mathcal{Y}$ and a set $U \subset \mathcal{X}$, we define $\text{err}(f, g|U)$ as the probability of disagreement between $f$ and $g$ conditioned on $\mathbf{x} \in U$, i.e., $\mathbb{P}(f(\mathbf{x}) \neq g(\mathbf{x}) \mid \mathbf{x} \in U)$. The probability is over $\mathbf{x} \sim \mathcal{D}$, which we often omit for simplicity. We focus on classifiers that minimize the error on the non-abstaining weak labels within the strong model hypothesis class $\mathcal{F}$, i.e., approximate solutions to $\arg\min_{f \in \mathcal{F}} \text{err}(f, f_{\text{weak}}|S)$. Our goal is to derive upper bounds on the error $\text{err}(f, y|D)$, which represents the classifier's error on the *true labels* on some $D \subset \mathcal{X}$.

**Definition 3** (Neighborhood). *(Lang et al., 2024) Let $\mathcal{N}$ be a* neighborhood function *that maps each point $x$ to a set of points $\mathcal{N}(x) \subset \mathcal{X}$ that we call the neighborhood of $x$. We will assume that $\mathcal{N}$ satisfies $x \in \mathcal{N}(x') \iff x' \in \mathcal{N}(x)$, i.e., that the neighborhoods are symmetric. We can extend $\mathcal{N}$ to a function of sets as $\mathcal{N}(A) = \bigcup_{x \in A} \mathcal{N}(x)$. Examples to keep in mind are $\mathcal{N}(x) = \{x' : \|\varphi(x) - \varphi(x')\| \leq r\}$ for some representation $\varphi : \mathcal{X} \to \mathbb{R}^d$, or, in the case of text inputs $x$, the set of fluent paraphrases of $x$. However, our results work with any definition of $\mathcal{N}$.*

**Definition 4** (Example graph). *(Lang et al., 2024) Let $G = (\mathcal{X}, E)$ be a graph with nodes representing elements of $\mathcal{X}$ (assumed to be finite but possibly very large), where two nodes $(x, x')$ are connected if $x \in \mathcal{N}(x')$ (or equivalently, $x' \in \mathcal{N}(x)$), with edge weight $w(x, x') = \mathbb{P}(x)\mathbb{P}(x')\mathbb{1}[x \in \mathcal{N}(x')]$.*

**Definition 5** ($\eta$-robust neighborhood size). *(Lang et al., 2024) For sets $A, U \subset \mathcal{X}$, the size of the $\eta$-robust neighborhood of $U$ in $A$ is: $P_{1-\eta}(U, A) := \min_{V \subset \mathcal{X}}\{\mathbb{P}(V|A) : w(V, U) \geq (1 - \eta)w(\mathcal{N}(U), U)\}$.*

**Definition 6** (Expansion). *(Lang et al., 2024) Fix sets $A, B \subset \mathcal{X}$. We say the distribution $\mathbb{P}_{\mathbf{x}}$ satisfies $(c, q)$-expansion on $(A, B)$ if for all sets $U \subset B$ with $\mathbb{P}(U|B) > q$, $\mathbb{P}(\mathcal{N}(U)|A) > c\mathbb{P}(U|B)$.*

**Definition 7** (Expansion of a set collection). *(Lang et al., 2024) A collection $\mathcal{M}$ of subsets of $B$ satisfies $(c, q)$-expansion on $(A, B)$ if for all $U \in \mathcal{M}$ with $\mathbb{P}(U|B) > q$, $\mathbb{P}(\mathcal{N}(U)|A) > c\mathbb{P}(U|B)$.*

**Definition 8** (Robust expansion). *(Lang et al., 2024) A collection $\mathcal{M}$ satisfies $(c, q, \eta)$-robust expansion on $(A, B)$ if for all $U \in \mathcal{M}$ with $P(U|B) > q$, $\mathbb{P}_{1-\eta}(U, A) > c\mathbb{P}(U|B)$. This recovers Definition 7 when $\eta = 0$.*

**Lemma D.1** (Lang et al. (2024)). *For a set $A \subset \mathcal{X}$ and classifier $f$, if $\mathbb{E}_{\mathbf{x} \sim \mathcal{D}|A, \mathbf{x}' \sim \mathcal{N}(\mathbf{x})}[f(\mathbf{x}) \neq f(\mathbf{x}')] \leq \gamma$ for some $\gamma > 0$, then for any $\eta > 0$, $\mathbb{P}(\overline{R_\eta(f)}|A) \leq \frac{\gamma}{\eta}$.*

**Good and bad edges.** (Lang et al., 2024) For a classifier $f$, a set $U \subset \mathcal{X}$, and $x' \in U, x \in \mathcal{N}(U)$, the pair $(x, x')$ is *bad* if $f(x) \neq f(x')$; otherwise, it is *good*. Let $\widetilde{\mathcal{N}}(U)$ be the subset of $\mathcal{N}(U)$ reachable by good edges. That is, $\widetilde{\mathcal{N}}(x') = \{x \in \mathcal{N}(x') : (x, x') \text{ good}\}$ and $\widetilde{\mathcal{N}}(A) = \cup_{x' \in A} \widetilde{\mathcal{N}}(x')$. The dependence of $\widetilde{\mathcal{N}}$ on $f$ is omitted for notational simplicity. If $f$ is $\eta$-robust on all points in $U$, then bad edges account for little of the weight between $\mathcal{N}(U)$ and $U$ in the example graph (Definition 4). Thus, if robust expansion is large and $f$ is $\eta$-robust on $U$, the neighborhood $\widetilde{\mathcal{N}}(U)$ reachable by good edges must also be large.

**Lemma D.2** (Lang et al. (2024)). *For any set $U \subset \mathcal{X}$ where $f$ satisfies $r(f, x) \leq \eta$ for all $x \in U$ (i.e., $U \subset R_\eta(f)$), we have: $w(\widetilde{\mathcal{N}}(U), U) \geq (1 - \eta)w(\mathcal{N}(U), U)$.*

**Expanding set family.** (Lang et al., 2024) Let $\mathcal{F}$ be the hypothesis class of the strong model, $y$ the ground-truth function, and $B \subset \mathcal{X}$. For each $f \in \mathcal{F}$, define the mistake set $U(B, f) = \{x \in B : f(x) \neq y(x)\}$. Then, the family of $\eta$-robust mistake sets is:

$$\mathcal{M}_\eta(B, \mathcal{F}) = \{R_\eta(f) \cap U(B, f) : f \in \mathcal{F}\}.$$

Similarly, define the family of $\eta$-robust non-mistake sets as:

$$\mathcal{M}'_\eta(B, \mathcal{F}) = \{R_\eta(f) \cap (B \setminus U(B, f)) : f \in \mathcal{F}\}.$$

**Problem Setup.** Suppose the input space can be partitioned into easy, hard, and overlapping points, i.e. $\mathcal{X} = D_{\text{easy only}} \cup D_{\text{hard only}} \cup D_{\text{overlap}}$. We assume that the pseudolabeler's accuracy is the same in the easy and overlapping regions ($D_{\text{easy only}}$ and $D_{\text{overlap}}$), and is higher in these regions compared to the hard region ($D_{\text{hard only}}$). Specifically, we have:

$$\varepsilon_1 = \mathbb{P}(y(\mathbf{x}) \neq f_{\text{weak}}(\mathbf{x})|S_i \cap D_{\text{easy only}}) = \mathbb{P}(y(\mathbf{x}) \neq f_{\text{weak}}(\mathbf{x})|S_i \cap D_{\text{overlap}})$$

$$\varepsilon_2 = \mathbb{P}(y(\mathbf{x}) \neq f_{\text{weak}}(\mathbf{x})|S_i \cap D_{\text{hard only}}) \geq \varepsilon_1$$

We assume that $0 < \varepsilon_1 \leq \varepsilon_2 \leq 0.5$. Further, denote the proportion of partitions as

$$p_i^{(\text{easy})} = \mathbb{P}(D_{\text{easy only}}|S_i)$$

$$p_i^{(\text{hard})} = \mathbb{P}(D_{\text{hard only}}|S_i)$$

$$p_i^{(\text{overlap})} = \mathbb{P}(D_{\text{overlap}}|S_i)$$

Our goal is to show how the overlap density can make strong model perform better in $p_i^{(\text{hard})}$ based on (robust) expansion property. Our main assumption is $\mathcal{M}'_\eta(S_i^{\text{good}} \cap D_{\text{overlap}}, \mathcal{F})$ satisfies $(c, q, \eta)$ robust expansion on $(S_i^{\text{bad}} \cap D_{\text{hard only}}, S_i^{\text{good}} \cap D_{\text{overlap}})$. It captures our intuition that strong model can learn something useful for hard data points from overlap data points.

In this setup, the trivial error bound obtained by perfectly mimicking the weak pseudolabeler is

$$\text{err}(f, y|S_i) = (p_i^{(\text{easy})} + p_i^{(\text{overlap})})\varepsilon_1 + p_i^{(\text{hard})}\varepsilon_2.$$

We claim that $\text{err}(f, y|S_i \cap D_{\text{hard only}}) < \varepsilon_2$ when the expansion coefficient $c$ — representing generalization from overlap density to hard density — is large, the model error rate in the overlap density is low, and the model exhibits sufficient robustness. Consequently, the large amount of weak-to-strong generalization can be attributed to $p_i^{(\text{hard})}(\varepsilon_2 - \text{err}(f, y|S_i \cap D_{\text{hard only}}))$, as observed in our synthetic experiments.

Our proof follows a similar structure to Lang et al. (2024).

Define

$$M_i = \{x \in S_i : f(x) \neq y(x)\}$$

$$E_i = \{x \in S_i : f(x) \neq f_{\text{weak}}(x)\}$$

$$U_i = S_i \backslash M_i$$

$$V_i = R_\eta(f) \cap U_i \cap S_i^{\text{good}} \cap D_{\text{overlap}}$$

Note that $V_i \subset U_i$. The following lemma shows that $V_i$ expands.

**Lemma D.3.** *Suppose an arbitrary classifier $f$ satisfies $\mathbb{P}(f(\mathbf{x}) \neq f_{weak}(x)$ or $f$ not $\eta$-robust at $\mathbf{x}|S_i \cap D_{overlap}) \leq 1 - q - \varepsilon_1$. Then, $\mathbb{P}(V_i|S_i^{good} \cap D_{overlap}) > q$.*

*Proof.* Suppose for a contradiction that $\mathbb{P}(V_i|S_i^{\text{good}} \cap D_{\text{overlap}}) \leq q$. Then by definition of $V_i$,

$$\mathbb{P}(V_i|S_i^{good} \cap D_{\text{overlap}}) = 1 - \mathbb{P}(\bar{V}_i|S_i^{\text{good}} \cap D_{\text{overlap}})$$

$$= 1 - \mathbb{P}(\bar{V}_i \cap S_i^{\text{good}} \cap D_{\text{overlap}}|S_i^{\text{good}} \cap D_{\text{overlap}})$$

$$= 1 - \mathbb{P}((S_i \backslash M_i) \cap R_\eta)^c \cap S_i^{\text{good}} \cap D_{\text{overlap}}|S_i^{\text{good}} \cap D_{\text{overlap}})$$

$$= 1 - \mathbb{P}((M_i \cap S_i^{\text{good}} \cap D_{\text{overlap}}) \cup R_\eta(f)^c|S_i^{\text{good}} \cap D_{\text{overlap}})$$

Fix an arbitrary $x \in M_i \cap S_i^{\text{good}} \cap D_{\text{overlap}}$. By definition of $M_i$, $f(x) \neq y(x)$. By definition of $S_i^{\text{good}}$, $f_{\text{weak}}(x) = y(x)$. Therefore, $f(x) \neq f_{\text{weak}}(x)$, thus $x \in E_i$. Since this holds for an arbitrary $x \in M_i \cap S_i^{\text{good}} \cap D_{\text{overlap}}$, $M_i \cap S_i^{\text{good}} \cap D_{\text{overlap}} \subset E_i$. Thus,

$$
\begin{aligned}
\mathbb{P}(V_i | S_i^{good} \cap D_{\text{overlap}}) &= 1 - \mathbb{P}((M_i \cap S_i^{\text{good}} \cap D_{\text{overlap}}) \cup R_\eta(f)^c | S_i^{\text{good}} \cap D_{\text{overlap}}) \\
&\geq 1 - \mathbb{P}(E_i \cup R_\eta(f)^c | S_i^{\text{good}} \cap D_{\text{overlap}}) \\
&= 1 - \frac{1}{1 - \varepsilon_1} \mathbb{P}(E_i \cup R_\eta(f)^c \cap S_i^{\text{good}} \cap D_{\text{overlap}} | S_i \cap D_{\text{overlap}}) \\
&\geq 1 - \frac{1}{1 - \varepsilon_1}(1 - q - \varepsilon_1) \qquad \because \text{ by assumption} \\
&= \frac{q}{1 - \varepsilon_1} \\
&> q \qquad \because \text{ since } 0 < \varepsilon_1 \leq 0.5
\end{aligned}
$$

$\square$

**Lemma D.4.** *Under the condition of Lemma D.3,*

$$
\mathbb{P}(U_i | S_i^{bad} \cap D_{hard\,only}) \geq c\mathbb{P}(V_i | S_i^{good} \cap D_{overlap})
$$

.

Since $\mathcal{M}'_\eta(S_i^{\text{good}} \cap D_{\text{overlap}}, \mathcal{F})$ satisfies $(c, q, \eta)$ robust expansion on $(S_i^{\text{bad}} \cap D_{\text{hard only}}, S_i^{\text{good}} \cap D_{overlap})$, we have

$$
P_{1-\eta}(V_i, S_i^{\text{bad}} \cap D_{\text{hard only}}) \geq c\mathbb{P}(V_i | S_i^{\text{good}} \cap D_{\text{overlap}})
$$

by the previous lemma. Also, by Lemma D.2,

$$
\mathbb{P}(\widetilde{\mathcal{N}}(V_i) | S_i^{\text{bad}} \cap D_{\text{hard only}}) \geq P_{1-\eta}(V_i, S_i^{\text{bad}} \cap D_{\text{hard only}})
$$

Fix an arbitrary $x \in \widetilde{\mathcal{N}}(V_i)$. By definition of $\widetilde{\mathcal{N}}(V_i)$, there exists $x' \in V_i$ such that $f(x) = f(x')$. Since $x' \in V_i \subset U_i$, $x' \in M_i$, thus we have that $f(x) = f(x') = y(x')$. And, $y(x) = y(x')$ since $x$ and $x'$ are both in $S_i$. This shows $f(x) = y(x)$, thus $x \notin M_i$, so $x \in U_i = S_i \backslash M_i$. Since $x$ was arbitrary, $\widetilde{\mathcal{N}}(V_i) \subset U_i$. This implies

$$
\mathbb{P}(U_i | S_i^{\text{bad}} \cap D_{\text{hard only}}) \geq \mathbb{P}(\widetilde{\mathcal{N}}(V_i) | S_i^{\text{bad}} \cap D_{\text{hard only}}),
$$

thus

$$
\mathbb{P}(U_i | S_i^{\text{bad}} \cap D_{\text{hard only}}) \geq c\mathbb{P}(V_i | S_i^{\text{good}} \cap D_{\text{overlap}}).
$$

We borrow the following lemma from Lang et al. (2024) directly.

**Lemma D.5** (Lang et al. (2024)). $E_i \supset (M_i \cap S_i^{good}) \cup (U_i \cap S_i^{bad})$.

The above lemma implies

$$
E_i \cap D_{\text{hard only}} \supset (M_i \cap S_i^{\text{good}} \cap D_{\text{hard only}}) \cup (U_i \cap S_i^{\text{bad}} \cap D_{\text{hard only}}).
$$

Now we state the formal version of Theorem 4.1 and provide the proof. Theorem 4.1 is obtained by setting $\eta = 0$, $q = 0$, thus $\mathbb{P}(R_\eta(f)^c | S_i^{good} \cap D_{\text{overlap}}) = 0$.

**Theorem D.1** (Formal version of Theorem 4.1). *Suppose $\mathcal{M}'_\eta(S_i^{good} \cap D_{overlap}, \mathcal{F})$ satisfies $(c, q, \eta)$-robust expansion on $(S_i^{bad} \cap D_{hard\,only}, S_i^{good} \cap D_{overlap})$ for some $c > 0$ and $\eta \geq 0$. Consider an arbitrary classifier $f$ such that $\mathbb{P}(f(\mathbf{x}) \neq f_{weak}(\mathbf{x})$ or $f$ not $\eta$-robust at $\mathbf{x} | S_i \cap D_{overlap}) \leq 1 - q - \varepsilon_1$. then $f$ satisfies the following error bound:*

$$
\begin{aligned}
err(f, y | S_i \cap D_{hard\,only}) \leq\ & err(f, f_{weak} | S_i \cap D_{hard\,only}) + \varepsilon_2 \\
& - 2c\varepsilon_2(1 - err(f, f_{weak} | S_i^{good} \cap D_{overlap}) - \mathbb{P}(R_\eta(f)^c | S_i^{good} \cap D_{overlap}))
\end{aligned}
$$

*Proof.* By Lemma D.5 and using $\mathbb{P}(S_i^{\text{good}}|S_i \cap D_{\text{hard only}}) = 1 - \varepsilon_2$, we have

$$\mathbb{P}(E_i|S_i \cap D_{\text{hard only}}) \geq P(M_i|S_i^{\text{good}} \cap D_{\text{hard only}})(1 - \varepsilon_2) + P(U_i|S_i^{\text{bad}} \cap D_{hard})\varepsilon_2$$

Applying Lemma D.4,

$$\mathbb{P}(E_i|S_i \cap D_{\text{hard only}}) \geq \mathbb{P}(M_i|S_i^{\text{good}} \cap D_{\text{hard only}})(1 - \varepsilon_2) + c\varepsilon_2\mathbb{P}(V_i|S_i^{\text{good}} \cap D_{\text{overlap}}) \qquad (1)$$

Applying Lemma D.4 again,

$$\begin{aligned}
\mathbb{P}(U_i|S_i \cap D_{\text{hard only}}) &= \varepsilon_2\mathbb{P}(U_i|S_i^{\text{bad}} \cap D_{\text{hard only}}) + (1 - \varepsilon_2)\mathbb{P}(U_i|S_i^{\text{good}} \cap D_{\text{hard only}}) \\
&\geq c\varepsilon_2\mathbb{P}(V_i|S_i^{\text{good}} \cap D_{\text{overlap}}) + (1 - \varepsilon_2)\mathbb{P}(U_i|S_i^{\text{good}} \cap D_{\text{hard only}})
\end{aligned}$$

then we have

$$\mathbb{P}(U_i|S_i^{\text{good}} \cap D_{\text{hard only}}) \leq \frac{1}{1 - \varepsilon_2}\left(\mathbb{P}(U_i|S_i \cap D_{\text{hard only}}) - c\varepsilon_2\mathbb{P}(V_i|S_i^{\text{good}} \cap D_{\text{overlap}})\right)$$

Combining this with $U_i = S_i \backslash M_i$,

$$\begin{aligned}
\mathbb{P}(M_i|S_i^{\text{good}} \cap D_{\text{hard only}}) &= 1 - \mathbb{P}(U_i|S_i^{\text{good}} \cap D_{\text{hard only}}) \\
&\geq 1 - \frac{1}{1 - \varepsilon_2}\left(\mathbb{P}(U_i|S_i \cap D_{\text{hard only}}) - c\varepsilon_2\mathbb{P}(V_i|S_i^{\text{good}} \cap D_{\text{overlap}})\right)
\end{aligned}$$

Plugging this into 1, we have

$$\begin{aligned}
\mathbb{P}(E_i|S_i \cap D_{\text{hard only}}) &\geq 1 - \varepsilon_2 - \mathbb{P}(U_i|S_i \cap D_{\text{hard only}}) + 2c\varepsilon_2\mathbb{P}(V_i|S_i^{\text{good}} \cap D_{\text{overlap}}) \\
&\geq 1 - \varepsilon_2 - (1 - \mathbb{P}(M_i|S_i \cap D_{\text{hard only}})) + 2c\varepsilon_2\mathbb{P}(V_i|S_i^{\text{good}} \cap D_{\text{overlap}}) \\
&= \mathbb{P}(M_i|S_i \cap D_{\text{hard only}})) - \varepsilon_2 + 2c\varepsilon_2\mathbb{P}(U_i \cap R_\eta(f)|S_i^{\text{good}} \cap D_{\text{overlap}}) \\
&= \mathbb{P}(M_i|S_i \cap D_{\text{hard only}})) - \varepsilon_2 + 2c\varepsilon_2\left(1 - \mathbb{P}(M_i \cup R_\eta(f)^c|S_i^{\text{good}} \cap D_{\text{overlap}})\right) \\
&\geq \mathbb{P}(M_i|S_i \cap D_{\text{hard only}})) + (2c - 1)\varepsilon_2 \\
&\quad - 2c\varepsilon_2\left(\mathbb{P}(M_i|S_i^{\text{good}} \cap D_{\text{overlap}}) + \mathbb{P}(R_\eta(f)^c|S_i^{\text{good}} \cap D_{\text{overlap}})\right)
\end{aligned}$$

Note that $\text{err}(f, f_{\text{weak}}|S_i \cap D_{\text{hard only}}) = \mathbb{P}(E_i|S_i \cap D_{\text{hard only}})$, and $\text{err}(f, y|S_i \cap D_{\text{hard only}}) = \mathbb{P}(M_i|S_i \cap D_{\text{hard only}})$. Plugging in those terms and rearranging leads to

$$\begin{aligned}
\text{err}(f, y|S_i \cap D_{\text{hard only}}) &\leq \text{err}(f, f_{\text{weak}}|S_i \cap D_{\text{hard only}}) + (1 - 2c)\varepsilon_2 \\
&\quad + 2c\varepsilon_2\text{err}(f, y|S_i^{good} \cap D_{\text{overlap}}) + 2c\varepsilon_2\mathbb{P}(R_\eta(f)^c|S_i^{good} \cap D_{\text{overlap}}) \\
&= \text{err}(f, f_{\text{weak}}|S_i \cap D_{\text{hard only}}) + (1 - 2c)\varepsilon_2 \\
&\quad + 2c\varepsilon_2\text{err}(f, f_{\text{weak}}|S_i^{good} \cap D_{\text{overlap}}) + 2c\varepsilon_2\mathbb{P}(R_\eta(f)^c|S_i^{good} \cap D_{\text{overlap}}) \\
&= \text{err}(f, f_{\text{weak}}|S_i \cap D_{\text{hard only}}) + \varepsilon_2 \\
&\quad - 2c\varepsilon_2(1 - \text{err}(f, f_{\text{weak}}|S_i^{good} \cap D_{\text{overlap}}) - \mathbb{P}(R_\eta(f)^c|S_i^{good} \cap D_{\text{overlap}}))
\end{aligned}$$

$\square$

## D.2 COVERAGE EXPANSION BY OVERLAP DENSITY

**Theorem D.2.** *Suppose $\mathcal{M}_\eta(T_i \cap D_{hard\,only}, \mathcal{F})$ satisfies $(c, q, \eta)$-robust expansion on $(S_i^{good}, T_i \cap D_{hard\,only})$ for some $c > 0$. Fix an arbitrary classifier $f : \mathcal{X} \to \mathcal{Y}$. The error of $f$ on $T_i \cap D_{hard\,only}$ is bounded by:*

$$err(f, y|T_i \cap \mathcal{D}_{hard}) \leq \mathbb{P}(R_\eta(f)^c | T_i \cap D_{hard\,only}) + \max\left(q, \frac{err(f, f_{weak}|S_i \cap D_{overlap})}{c(1 - \varepsilon_1)}\right)$$

*Proof.* Again, the proof follows the similar steps to Lang et al. (2024). Define $M_i = \{x : f(x) \neq y(x)\} \cap T_i \cap D_{hard\,only}$ as the set of mistakes of $f$ in $T_i \cap D_{hard\,only}$, and let $U_i = M_i \cap R_\eta(f)$. Let $E_i = \{x \in S_i \cap D_{overlap}\}$ be the set of points in $S_i \cap \mathcal{D}_{overlap}$ where $f$ disagrees with the weak labels. Since we have $err(f, f_{weak}|S_i \cap D_{overlap}) = \mathbb{P}(E_i | S_i)$ and $err(f, y|T_i \cap D_{hard\,only}) = \mathbb{P}(M_i | T_i \cap D_{hard\,only}) \leq \mathbb{P}(U_i | T_i \cap D_{hard\,only}) + \mathbb{P}(R_\eta(f)^c | T_i \cap D_{hard\,only})$ by union bound, it suffices to bound $\mathbb{P}(R_\eta(f)^c | T_i \cap D_{hard\,only})$. Since $U_i \subset R_\eta(f)$, we have $\mathbb{P}(\tilde{\mathcal{N}}(U_i) | S_i^{good} \cap D_{overlap}) \geq P_{1-\eta}(U_i, S_i^{good} \cap D_{overlap})$ by Lemma D.2. Also, $U_i \in \mathcal{M}_\eta(T_i, \mathcal{F})$ by definition. Then, $\mathbb{P}(U_i | T_i \cap D_{hard\,only}) > q$ and $(c, q, \eta)$-robust expansion implies

$$\mathbb{P}(\tilde{\mathcal{N}}(U_i) | S_i^{good} \cap D_{overlap}) \geq P_{1-\eta}(U_i, S_i^{good} \cap D_{overlap}) > c\mathbb{P}(U_i | T_i \cap D_{hard\,only}).$$

We proceed in two cases.

**Case 1:** $\mathbb{P}(U_i | T_i \cap D_{hard\,only}) \leq q$. In this case, we directly obtain $err(f, y|T_i \cap D_{hard\,only}) = \mathbb{P}(M_i | T_i \cap D_{hard\,only}) \leq \mathbb{P}(R_\eta(f)^c | T_i \cap D_{hard\,only}) + \mathbb{P}(U_i | T_i \cap D_{hard\,only}) \leq \mathbb{P}(R_\eta(f)^c | T_i \cap D_{hard\,only}) + q$.

**Case 2:** $\mathbb{P}(U_i | T_i \cap D_{hard\,only}) > q$. In this case, by the assumption that $(S_i^{good} \cap D_{overlap}, T_i \cap D_{hard\,only})$ satisfy $(c, q, \eta)$-robust expansion and $\mathbb{P}\tilde{\mathcal{N}}(U_i) | S_i^{good} \cap D_{overlap}) > c\mathbb{P}(U_i | T_i \cap D_{hard\,only})$, we have

$$\mathbb{P}(\tilde{\mathcal{N}}(U_i) \cap S_i^{good} \cap D_{overlap} | S_i \cap D_{overlap}) = (1 - \varepsilon_1)\mathbb{P}(\tilde{\mathcal{N}}(U_i) \cap | S_i^{good} \cap D_{overlap})$$
$$\geq (1 - \varepsilon_1)c\mathbb{P}(U_i | T_i \cap D_{hard\,only}).$$

Suppose $x \in \tilde{\mathcal{N}}(U_i) \cap S_i^{good} \cap D_{overlap}$. By the definition of $\tilde{\mathcal{N}}(U_i)$, there exists a point $x' \in U_i$ reachable from x by a good edge, such that $f(x) = f(x')$. Then, since $x \in S_i^{good}$, $f_{weak}(x) = y(x) = y(x')$. Also, since $x' \in M_i$, $y(x') \neq f(x') = f(x)$. Thus, $f(x) \neq f_{weak}(x)$, which implies $x \in E_i$. This leads to

$$err(f, f_{weak}|S_i \cap D_{overlap}) = \mathbb{P}(E_i | S_i \cap D_{overlap}) \geq \mathbb{P}(\tilde{\mathcal{N}}(U_i) \cap S_i^{good} | S_i) \geq c(1-\varepsilon_1)\mathbb{P}(U_i | T_i \cap D_{hard\,only}),$$

Rearranging the inequality, we have

$$err(f, y|T_i \cap \mathcal{D}_{hard}) \leq \mathbb{P}(R_\eta(f)^c | T_i \cap D_{hard\,only}) + \frac{err(f, f_{weak}|S_i \cap D_{overlap})}{c(1 - \varepsilon_1)}$$

$\square$

## D.3 PROOF OF THEOREM 4.2

**Setup** We extend the setup in Section 3. We consider label-conditioned Gaussian mixtures as follows. We denote mean parameters as $\mu_{overlap} = \begin{bmatrix} \tilde{\mu}_{easy} \\ \tilde{\mu}_{hard} \end{bmatrix}$, where $\tilde{\mu}_{easy} \in \mathbb{R}^{d_{easy}}$ and $\tilde{\mu}_{hard} \in \mathbb{R}^{d_{hard}}$. We set $\mu_{easy} = \begin{bmatrix} \tilde{\mu}_{easy} \\ 0 \end{bmatrix}$, $\mu_{hard} = \begin{bmatrix} 0 \\ \tilde{\mu}_{hard} \end{bmatrix}$, to instantiate the distribution of overlap, easy-only and hard-only data points. We set up a common covariance $\Sigma = c \begin{bmatrix} I_{easy} & 0 \\ 0 & I_{hard} \end{bmatrix}$ where $I_{easy}$ and $I_{hard}$ are identity matrices. The distribution of input $\mathbf{x}$ follows Gaussian mixtures

$$\mathbb{P}(\mathbf{x}|\mathbf{y} = 1) = \pi_{easy}\mathbf{N}(\mu_{easy}, \Sigma) + \pi_{hard}\mathbf{N}(\mu_{hard}, \Sigma) + \pi_{overlap}\mathbf{N}(\mu_{overlap}, \Sigma)$$

$$\mathbb{P}(\mathbf{x}|\mathbf{y}=-1) = \pi_{\text{easy}}\mathbf{N}(-\mu_{\text{easy}},\Sigma) + \pi_{\text{hard}}\mathbf{N}(-\mu_{\text{hard}},\Sigma) + \pi_{\text{overlap}}\mathbf{N}(-\mu_{\text{overlap}},\Sigma),$$

where $\pi_{\text{easy}} \geq 0, \pi_{\text{hard}} \geq 0, \pi_{\text{overlap}} \geq 0, \pi_{\text{easy}} + \pi_{\text{hard}} + \pi_{\text{overlap}} = 1$. Assuming $\mathbb{P}(\mathbf{y}=1) = \mathbb{P}(\mathbf{y}=-1) = 0.5$, $D_{\text{easy only}}, D_{\text{hard only}}, D_{\text{overlap only}}$ can be described as

$$D_{\text{easy only}} \sim \frac{1}{2}\mathbf{N}(-\mu_{\text{easy}},\Sigma) + \frac{1}{2}\mathbf{N}(\mu_{\text{easy}},\Sigma)$$

$$D_{\text{hard only}} \sim \frac{1}{2}\mathbf{N}(-\mu_{\text{hard}},\Sigma) + \frac{1}{2}\mathbf{N}(\mu_{\text{hard}},\Sigma)$$

$$D_{\text{overlap}} \sim \frac{1}{2}\mathbf{N}(-\mu_{\text{overlap}},\Sigma) + \frac{1}{2}\mathbf{N}(\mu_{\text{overlap}},\Sigma)$$

**Lemma D.6.** $\frac{1}{\sqrt{1-y}} \leq e^{\frac{y}{2(1-y)}}$ for $0 < y < 1$.

*Proof.* Consider function $f(y) = \ln(\frac{1}{\sqrt{1-y}}) - \frac{y}{2(1-y)} = -\frac{1}{2}\ln(1-y) - \frac{y}{2(1-y)}$. It suffices to show $f(y) \leq 0$, which implies $\frac{1}{\sqrt{1-y}} \leq e^{\frac{y}{2(1-y)}}$ by taking the exponential. First, we can derive

$$f'(y) = \frac{1}{2(1-y)} - \frac{1}{2(1-y)^2}$$
$$= -\frac{y}{2(1-y)^2}$$

Thus, we can see $f'(y) < 0$ for $0 < y < 1$. Also, we have $f(0) = 0$. Then, since $f(0) = 0$ and $f$ is decreasing for $0 < y < 1$, $f(y) \leq 0$. $\qquad\square$

**Lemma D.7.** *Suppose $X_1 \sim N(\mu_1,\sigma_1^2)$, $X_2 \sim N(\mu_2,\sigma_2^2)$. Then, $X_1X_2 - \mu_1\mu_2 \sim SE(\nu^2,b)$, where $SE$ represents a subexponential with parameters $\nu^2 = \mu_1^2\sigma_2^2 + \mu_2^2\sigma_1^2 + \frac{4}{3}\sigma_1^2\sigma_2^2$, $b = \frac{1}{2\sigma_1\sigma_2}$.*

*Proof.* We can write
$$X_1 = \mu_1 + \sigma_1 Z_1$$
$$X_2 = \mu_2 + \sigma_2 Z_2,$$

where $Z_1, Z_2 \sim N(0,1)$. Then, $X_1X_2 - \mu_1\mu_2 = \mu_1\sigma_2 Z_2 + \mu_2\sigma_1 Z_1 + \sigma_1\sigma_2 Z_1 Z_2$. Let $A = \mu_1\sigma_2 Z_2 + \mu_2\sigma_1 Z_1$. Since $A \sim N(0, \mu_1^2\sigma_2^2 + \mu_2^2\sigma_1^2)$, $A$ is subgaussian with variance proxy $\sigma_A^2 = \mu_1^2\sigma_2^2 + \mu_2^2\sigma_1^2$. This implies $A \sim SE(\nu_A^2, b_A)$ for any $b_A > 0$, where $\nu_A^2 = \mu_1^2\sigma_2^2 + \mu_2^2\sigma_1^2$. We choose $b_A = 2\sigma_1\sigma_2$.

Next, let $B = \sigma_1\sigma_2 Z_1 Z_2$. We can obtain MGF of $B$ as

$$\mathbb{E}[e^{\lambda B}] = \frac{1}{\sqrt{1-(\lambda\sigma_1\sigma_2)^2}}, \text{ for } |\lambda| < \frac{1}{\sigma_1\sigma_2}.$$

Especially, for $|\lambda| < \frac{1}{2\sigma_1\sigma_2}$, we can bound the MGF using the inequality in Lemma D.6 with $y = (\lambda\sigma_1\sigma_2)^2$. Thus,

$$\mathbb{E}[e^{\lambda B}] \leq \exp\left(\frac{(\lambda\sigma_1\sigma_2)^2}{2(1-(\lambda\sigma_1\sigma_2)^2)}\right).$$

For $|\lambda| < \frac{1}{2\sigma_1\sigma_2}$, we have $(\lambda\sigma_1\sigma_2)^2 < \frac{1}{4}$, so $1 - (\lambda\sigma_1\sigma_2)^2 > \frac{3}{4}$. Therefore,

$$\mathbb{E}[e^{\lambda X}] \leq \exp\left(\frac{2}{3}(\lambda\sigma_1\sigma_2)^2\right).$$

By comparing it with the MGF bound for any subexponential random variable $Y$,

$$\mathbb{E}[e^{\lambda Y}] \leq \exp\left(\frac{\lambda^2\nu^2}{2}\right), \text{ for } |\lambda| < \frac{1}{b},$$

we identify the subexponential parameters for $B$ as $\nu_B^2 = \frac{4}{3}(\sigma_1\sigma_2)^2$, $b_B = 2\sigma_1\sigma_2$. Since $A \sim SE(\nu_A^2, b_A), B \sim SE(\nu_B^2, b_B)$, we have

$$X_1X_2 - \mu_1\mu_2 \sim SE\left(\nu_A^2 + \nu_B^2, \max(b_A, b_B)\right)$$

by the additivity of subexponential. Thus, we have

$$X_1X_2 - \mu_1\mu_2 \sim SE(\mu_1^2\sigma_2^2 + \mu_2^2\sigma_1^2 + \frac{4}{3}\sigma_1^2\sigma_2^2, 2\sigma_1\sigma_2)$$

$\qquad\square$

*Proof of Theorem 4.2.* Let $\mathbf{x}_{\text{diff}} = \mathbf{x}_{\text{overlap}} - \mathbf{x}_{\text{easy only}}$ so that

$$\mathbb{E}[\mathbf{x}_{\text{overlap}}{}^\top \mathbf{x}_{\text{hard only}}] - \mathbb{E}[\mathbf{x}_{\text{easy only}}{}^\top \mathbf{x}_{\text{hard only}}] = \mathbb{E}[\mathbf{x}_{\text{diff}}^\top \mathbf{x}_{\text{hard only}}].$$

This difference distribution will follow $\mathbf{x}_{\text{diff}} \sim \mathbf{N}(\mu_{\text{overlap}} - \mu_{\text{easy only}}, 2cI) = \mathbf{N}(\mu_{\text{hard}}, 2cI)$. Let $\mathbf{z} = (\mathbf{x}_{\text{diff}})_i (\mathbf{x}_{\text{hard only}})_i$. Then, $\mathbf{x}_{\text{easy only}}^\top \mathbf{x}_{\text{hard only}} = \sum_{i=1}^d z_i$. Since

$$(\mathbf{x}_{\text{diff}})_i \sim \mathbf{N}((\mu_{\text{hard}})_i, 2c)$$

$$(\mathbf{x}_{\text{hard only}})_i \sim \mathbf{N}((\mu_{\text{hard}})_i, c),$$

we have $\mathbf{z}_i \sim SE\left(3c(\mu_{\text{hard}})_i^2 + \frac{8}{3}c^2, 4c\right)$ by Lemma D.7. By additivity,

$$\mathbf{x}_{\text{diff}}^\top \mathbf{x}_{\text{hard only}} = \sum_{i=1}^d \mathbf{z}_i \sim SE\left(\frac{8}{3}dc^2 + 3c\|\mu_{\text{hard}}\|_2^2, 4c\right).$$

The concentration inequality then follows directly from the standard concentration inequality for subexponential distributions (Wainwright, 2019). We have

$$\mathbb{P}\left(\mathbf{x}_{\text{overlap}}^\top \mathbf{x}_{\text{hard only}} - \mathbf{x}_{\text{easy only}}^\top \mathbf{x}_{\text{hard only}} \leq \|\mu_{\text{hard}}\|_2^2 - t\right) \leq \exp\left(-\min(\frac{t^2}{2\nu^2}, \frac{t}{2b})\right),$$

where $\nu^2 = \frac{8}{3}dc^2 + 3c\|\mu_{\text{hard}}\|_2^2$ and $b = 4c$. By plugging in $t = \|\mu_{\text{hard}}\|_2^2$, we can obtain the average error rate bound

$$\mathbb{P}\left(\mathbf{x}_{\text{overlap}}^\top \mathbf{x}_{\text{hard only}} \leq \mathbf{x}_{\text{easy only}}^\top \mathbf{x}_{\text{hard only}}\right) \leq \exp\left(-\min\left(\frac{\|\mu_{\text{hard}}\|_2^4}{\frac{16}{3}dc^2 + 6c\|\mu_{\text{hard}}\|_2^2}, \frac{\|\mu_{\text{hard}}\|_2^2}{8c}\right)\right).$$

$\square$

## D.4 PROOF OF THEOREM 4.3

*Proof.* Recall that $o(s) = \mathbb{P}(\mathcal{D}_{\text{overlap}}|\mathcal{D}_s)$ is the population overlap density of data source $s$, $\bar{o}_t = \frac{|\bar{V}_t(s)|}{|\bar{D}_t(s)|}$ is the empirical overlap density at round $t$, and $o^* = \max_s o(s)$ is the optimal overlap density. Define $r_t(s) = \sqrt{\frac{2\log T}{n_t(s)}}$, where $n_t(s)$ denotes the number of times data source $s$ has been chosen up to round t. We first show that

$$\mathbb{P}\left(|\bar{o}_t(s) - o(s)| \geq r_t(s)\right) \leq 2T^{-2}.$$

Here, $t$ is a random variable, thus we cannot apply Hoeffding's inequality directly. Instead, we replace $t$ with some fixed round $j$ first. Define $\bar{v}_j(s) = \frac{|\bar{V}_j(s)|}{|\bar{D}_j(s)|}$, which represents the average overlap ratio at the data source $D_s$ from the first $j$ rounds ($j \leq T$). Since $j$ is fixed and $0 \leq \bar{v}_j(s) \leq 1$, by Hoeffding's inequality, we obtain

$$\mathbb{P}\left(|\bar{v}_j(s) - o(s)| \geq r_j(s)\right) \leq 2T^{-4}.$$

Define the event $\mathcal{E} = \{\forall s \in [K], \forall j \leq T, |\bar{v}_j(s) - o(s)| \geq r_j(s)\}$. Then, by the union bound, we have

$$\mathbb{P}[\mathcal{E}] \leq \sum_{s=1}^K \sum_{j=1}^T \mathbb{P}\left(|\bar{v}_j(s) - o(s)| \geq r_j(s)\right) \leq \sum_{s=1}^K \sum_{j=1}^T 2T^{-4} = 2KT^{-3} \leq 2T^{-2}.$$

Thus, for the random variable $n_t(s)$, it holds that

$$\mathbb{P}\left(|\bar{o}_t(s) - o(s)| \geq r_t(s)\right) \leq 2T^{-2}.$$

Define the regret associated with the choice of source $D_s$ as $\Delta(s) := o^* - o(s)$ and the contribution of data source $D_s$ to the accumulated regret at round $t$ as $R(t, s) = n_t(s)\Delta(s)$. The total regret is then given by $R(t) = \sum_{s=1}^K R(t, s) = t(o^* - \bar{o}_t)$.

Suppose $|\bar{o}_t(s) - o(s)| \geq r_t(s)$, which is an event in $\mathcal{E}^c$. According to the algorithm, $\text{UCB}_t(s_t) \geq \text{UCB}_t(s^*)$, where $s^*$ denotes the index of the optimal source and where $s_t$ denotes the data source chosen at round $t$. Trivially, we also have $\text{UCB}_t(s^*) \geq o(s^*)$. Thus, we have

$$o(s_t) + 2r_t(s_t) \geq \bar{o}_t(s_t) + r_t(s_t) = \text{UCB}_t(s_t) \geq \text{UCB}_t(s^*) \geq o(s^*),$$

which implies

$$\Delta(s_t) = o(s^*) - o(s_t) \leq 2r_t(s_t) = 2\sqrt{\frac{2\log T}{n_t(s_t)}}.$$

From this, we can obtain

$$
\begin{aligned}
R(t) &= \sum_{s=1}^{K} R(t, s) \\
&= \sum_{s=1}^{K} n_t(s)\Delta(s) \\
&= \sum_{s=1}^{K} n_t(s) 2\sqrt{2\log T / n_t(s)} \\
&= \sum_{s=1}^{K} 2\sqrt{2n_t(s)\log T} \\
&= 2K\sqrt{\log T} \sum_{s=1}^{K} \frac{1}{K}\sqrt{2n_t(s)} \\
&\leq 2K\sqrt{\log T} \sqrt{\frac{1}{K}\sum_{s=1}^{K} n_t(s)} \qquad \because \text{Jensen's inequality} \\
&= 2K\sqrt{\log T}\sqrt{\frac{t}{K}} \\
&= 2\sqrt{Kt\log T}
\end{aligned}
$$

We have the final result from this.

$$
\begin{aligned}
\mathbb{E}[R(t)] &= \mathbb{P}(\mathcal{E})E[R(t)|\mathcal{E}] + \mathbb{P}(\mathcal{E}^c)E[R(t)|\mathcal{E}^c] \\
&\leq 2T^{-2} * T + \mathbb{P}(\mathcal{E}^c)E[R(t)|\mathcal{E}^c] \qquad \because R(T) \leq T \text{ trivially} \\
&\leq 2T^{-1} + (1 - 2T^{-2})2\sqrt{Kt\log T} \qquad \because \text{plugging in the previous result} \\
&= O(\sqrt{Kt\log T})
\end{aligned}
$$

It follows that

$$\mathbb{E}[o^* - \bar{o}_t] = \mathbb{E}[R(T)/t] \leq O\left(\sqrt{\frac{K\log T}{t}}\right).$$

$\square$

# E  EXPERIMENT DETAILS

**Real datasets.** In our language model experiments, we used a subset of datasets in EleutherAI (2021), which includes ANLI-R2 (Nie et al., 2020), CoLA (Warstadt et al., 2018), DREAM (Sun et al., 2019), MC-TACO (Ben Zhou & Roth, 2019), HelleSwag (Zellers et al., 2019), MultiRC (Khashabi et al., 2018), PAWS (Zhang et al., 2019), PICa (Yang et al., 2022), QuAIL (Rogers et al., 2020), QUARTZ (Tafjord et al., "2019"), Social IQa Sap et al. (2019), SST2 (Socher et al., 2013), WiC(Pilehvar & Camacho-Collados, 2019), Tweet Sentiment Naji (2012), Anthropic HH-RLHF (Ganguli et al., 2022), SciQ (Welbl et al., 2017), CosmosQA (Huang et al., 2019), BoolQ (Clark et al.,

2019), and the Amazon Polarity (Zhang et al., 2015) datasets. The training dataset was divided into the weak model training data, $D_{\text{train}}$, and the weak-to-strong model training data, $D_{\text{w2s}}$. We sampled $n_{\text{train}} = 10,000$, $n_{\text{val}} = 1,000$, and $n_{\text{test}} = 5,000$ for the training, validation, and test datasets, respectively, for datasets with larger splits than the specified sizes, in accordance with the default parameters provided in https://github.com/EleutherAI/w2s.

In Wrench experiments, we used a subset of Wrench benchmark(Zhang et al., 2021), which includes CDR, Census, Commercial, iMDb, Mushroom, SMS, Spambase, Tennis, Yelp, and Youtube datasets.

**Dataset sizes**   Table A2, A3 show dataset sizes, the sizes of detected easy-only, hard-only, and overlap sizes, and $n_{\text{controlled}}$ that is used for overlap mechanism experiments in Section 5.1.

Table A2: Summary of data statistics from Section 5.3.1. The mean and standard deviation are calculated from 20 repeated experiments, each using different random seeds.

| Dataset | $n_{\text{train}}$ | $n_{\text{test}}$ | $n_{\text{w2s}}$ | $n_{\text{controlled}}$ | $|\bar{D}_{\text{easy only}}|$ | $|\bar{D}_{\text{hard only}}|$ | $|\bar{D}_{\text{overlap}}|$ | Sample size per round |
|---|---|---|---|---|---|---|---|---|
| amazon_polarity | 5000 | 5000 | 5000 | $1150 \pm 105$ | $744 \pm 94$ | $407 \pm 29$ | $3850 \pm 105$ | $73 \pm 3$ |
| anli-r2 | 5000 | 668 | 5000 | $1404 \pm 45$ | $1734 \pm 40$ | $1863 \pm 39$ | $1404 \pm 45$ | $72 \pm 2$ |
| anthropic_hh | 5000 | 5000 | 5000 | $1976 \pm 62$ | $296 \pm 13$ | $2729 \pm 69$ | $1976 \pm 62$ | $54 \pm 2$ |
| boolq | 3053 | 2474 | 3053 | $1483 \pm 27$ | $595 \pm 35$ | $922 \pm 40$ | $1537 \pm 52$ | $56 \pm 1$ |
| cola | 2028 | 644 | 2028 | $568 \pm 45$ | $126 \pm 34$ | $442 \pm 21$ | $1461 \pm 45$ | $27 \pm 1$ |
| cosmos_qa | 5000 | 2646 | 5000 | $2376 \pm 62$ | $615 \pm 84$ | $1994 \pm 46$ | $2392 \pm 89$ | $78 \pm 2$ |
| dream | 2541 | 1984 | 2541 | $1037 \pm 45$ | $261 \pm 13$ | $777 \pm 42$ | $1504 \pm 45$ | $43 \pm 1$ |
| hellaswag | 5000 | 5000 | 5000 | $1545 \pm 52$ | $499 \pm 32$ | $2957 \pm 49$ | $1545 \pm 52$ | $39 \pm 2$ |
| mc_taco | 2698 | 2458 | 2698 | $1250 \pm 61$ | $756 \pm 50$ | $693 \pm 43$ | $1250 \pm 61$ | $49 \pm 2$ |
| multirc | 5000 | 4150 | 5000 | $1763 \pm 57$ | $1267 \pm 70$ | $1970 \pm 31$ | $1763 \pm 57$ | $66 \pm 3$ |
| paws | 5000 | 5000 | 5000 | $663 \pm 43$ | $1808 \pm 57$ | $2529 \pm 42$ | $663 \pm 43$ | $51 \pm 1$ |
| piqa | 5000 | 1832 | 5000 | $1591 \pm 51$ | $627 \pm 33$ | $2783 \pm 26$ | $1591 \pm 51$ | $59 \pm 1$ |
| quail | 4613 | 2148 | 4613 | $1724 \pm 61$ | $890 \pm 65$ | $2000 \pm 43$ | $1724 \pm 61$ | $60 \pm 2$ |
| quartz | 833 | 764 | 833 | $347 \pm 34$ | $302 \pm 37$ | $185 \pm 13$ | $347 \pm 34$ | $12 \pm 1$ |
| sciq | 4837 | 2980 | 4837 | $2240 \pm 44$ | $831 \pm 49$ | $1767 \pm 53$ | $2240 \pm 44$ | $82 \pm 2$ |
| social_i_qa | 5000 | 1888 | 5000 | $1578 \pm 51$ | $998 \pm 40$ | $2424 \pm 36$ | $1578 \pm 51$ | $60 \pm 2$ |
| sst2 | 5000 | 856 | 5000 | $1876 \pm 48$ | $984 \pm 49$ | $892 \pm 16$ | $3125 \pm 48$ | $95 \pm 2$ |
| twitter-sentiment | 5000 | 5000 | 5000 | $2173 \pm 69$ | $880 \pm 36$ | $1293 \pm 50$ | $2827 \pm 69$ | $98 \pm 1$ |
| wic | 2214 | 638 | 2214 | $1073 \pm 31$ | $325 \pm 36$ | $813 \pm 34$ | $1076 \pm 34$ | $37 \pm 1$ |

Table A3: Summary of data statistics from Section 5.3.1. The mean and standard deviation are calculated from 20 repeated experiments, each using different random seeds.

| Dataset | $n_{\text{train}}$ | $n_{\text{test}}$ | $n_{w2s}$ | $n_{\text{controlled}}$ | $|\bar{D}_{\text{easy only}}|$ | $|\bar{D}_{\text{hard only}}|$ | $|\bar{D}_{\text{overlap}}|$ |
|---|---|---|---|---|---|---|---|
| cdr | 4215 | 4673 | 4215 | $1108 \pm 63$ | $191 \pm 37$ | $2916 \pm 67$ | $1108 \pm 63$ |
| census | 5041 | 16281 | 5042 | $1629 \pm 66$ | $280 \pm 69$ | $1349 \pm 30$ | $3413 \pm 66$ |
| commercial | 32065 | 7496 | 32065 | $8988 \pm 934$ | $2216 \pm 835$ | $6773 \pm 616$ | $23077 \pm 934$ |
| imdb | 10000 | 2500 | 10000 | $2469 \pm 688$ | $630 \pm 425$ | $4885 \pm 3004$ | $4485 \pm 2621$ |
| sms | 2285 | 500 | 2286 | $662 \pm 107$ | $117 \pm 39$ | $1507 \pm 113$ | $662 \pm 107$ |
| spambase | 1840 | 461 | 1840 | $633 \pm 34$ | $102 \pm 26$ | $531 \pm 24$ | $1207 \pm 34$ |
| tennis | 3479 | 1098 | 3480 | $754 \pm 688$ | $680 \pm 706$ | $77 \pm 40$ | $2723 \pm 693$ |
| yelp | 15200 | 3800 | 15200 | $5721 \pm 485$ | $1160 \pm 414$ | $4934 \pm 1387$ | $9106 \pm 1246$ |
| youtube | 793 | 250 | 793 | $341 \pm 28$ | $54 \pm 21$ | $290 \pm 24$ | $449 \pm 34$ |

**Synthetic datasets.** Synthetic datasets in Section 5.3 are generated with Gaussian mixture distribution. We first sample mean parameters $\mu_{\text{overlap}} = \begin{bmatrix} \tilde{\mu}_{\text{easy}} \\ \tilde{\mu}_{\text{hard}} \end{bmatrix}$ from uniform distribution, where

$\tilde{\mu}_{\text{easy}} \in \mathbb{R}^{d_{\text{easy}}}$ and $\tilde{\mu}_{\text{hard}} \in \mathbb{R}^{d_{\text{hard}}}$. We set $\mu_{\text{easy}} = \begin{bmatrix} \tilde{\mu}_{\text{easy}} \\ 0 \end{bmatrix}$, $\mu_{\text{hard}} = \begin{bmatrix} 0 \\ \tilde{\mu}_{\text{hard}} \end{bmatrix}$, to simulate data points with easy+hard (overlap), easy, hard patterns. Similarly, we set up the covariances $\Sigma_{\text{easy}} = \Sigma_{\text{hard}} = \Sigma_{\text{overlap}} = c \begin{bmatrix} I_{\text{easy}} & 0 \\ 0 & I_{\text{hard}} \end{bmatrix}$, $I_{\text{easy}}$ and $I_{\text{hard}}$ being identity matrices. We used $c = 5$, $d_{\text{easy}} = d_{\text{hard}} = 20$ in synthetic experiments. Labels (Y) are sampled from $\{-1, 1\}$ uniformly. The distribution of input $X$ follows Gaussian mixtures

$$P(X|Y = 1) \sim \pi_{\text{easy}} \mathbf{N}(\mu_{\text{easy}}, \Sigma_{\text{easy}}) + \pi_{\text{hard}} \mathbf{N}(\mu_{\text{hard}}, \Sigma_{\text{hard}}) + \pi_{\text{overlap}} \mathbf{N}(\mu_{\text{overlap}}, \Sigma_{\text{overlap}})$$

$$P(X|Y = -1) \sim \pi_{\text{easy}} \mathbf{N}(-\mu_{\text{easy}}, \Sigma_{\text{easy}}) + \pi_{\text{hard}} \mathbf{N}(-\mu_{\text{hard}}, \Sigma_{\text{hard}}) + \pi_{\text{overlap}} \mathbf{N}(-\mu_{\text{overlap}}, \Sigma_{\text{overlap}}),$$

where $\pi_{\text{easy}} \geq 0, \pi_{\text{hard}} \geq 0, \pi_{\text{overlap}} \geq 0$, $\pi_{\text{easy}} + \pi_{\text{hard}} + \pi_{\text{overlap}} = 1$. We control parameters $\pi_{\text{easy}}, \pi_{\text{hard}}, \pi_{\text{overlap}}$ to see how they affect the weak to strong generalization.

We fixed $n_{\text{easy}} = n_{\text{hard}} = 100$ and increase 5 overlap data points each time. We trained weak-to-strong model on overlap data points only.

**Computing resources** We used a GPU cluster with 8 NVIDIA A100 SXM2 40GB HBM2 NV-LINK, 2x Intel® Xeon Cascade Lake 5218 (2.3GHz) Processor (24-Core), 16x 32 GB ECC REG DDR4-2933 RAM.

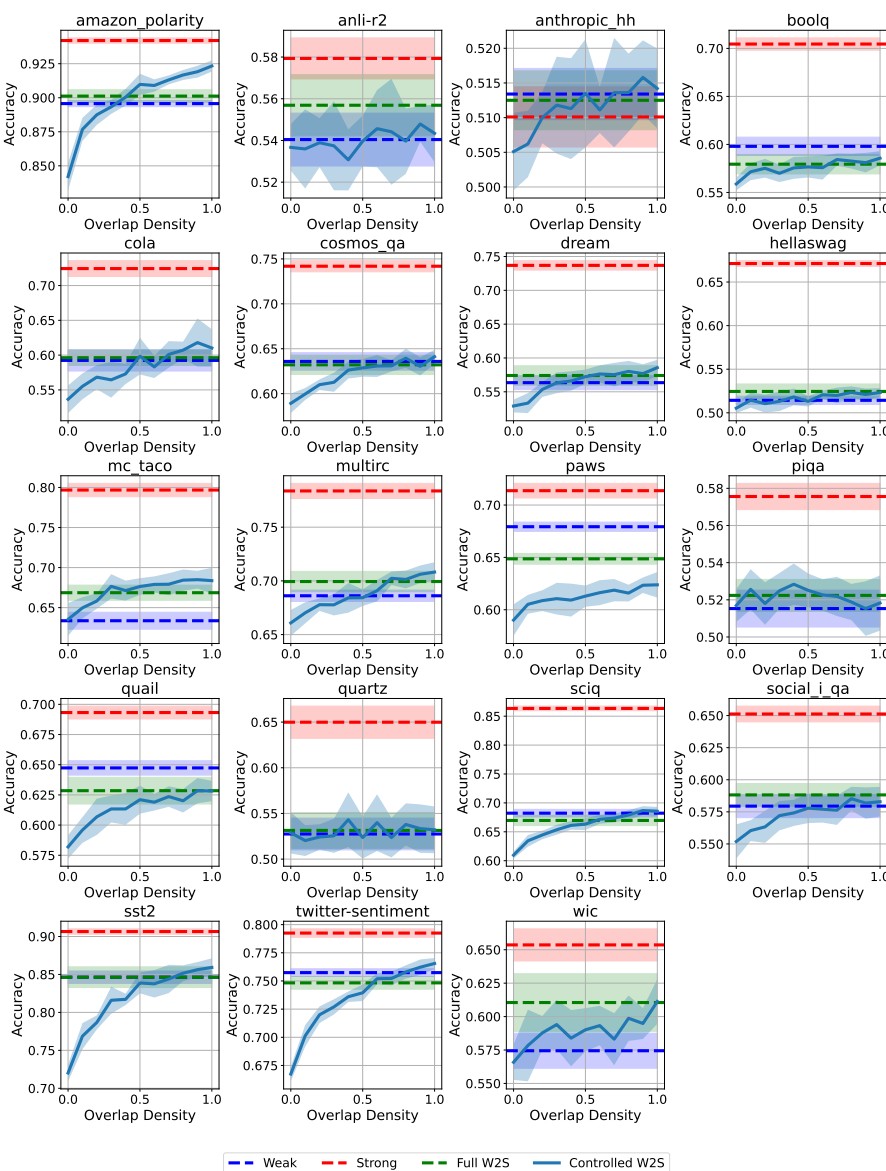

Figure A1: Overlap density versus performance in weak-to-hard generalization with large language models. The red lines represent the accuracies of strong models trained on true labels, while the blue dashed lines indicate the accuracies of weak models on the test set. The green dashed lines (Full W2S) show the accuracies of weak-to-strong models trained on the entire pseudolabeled dataset. Lastly, the Controlled W2S lines represent the accuracies of strong models trained on data with a controlled proportion of overlap density. In general, the strong model's improvement over the weak model tracks the overlap proportion, suggesting that the overlap density is indeed an important mechanism for generalization.

# F  ADDITIONAL EXPERIMENTAL RESULTS

## F.1  COMPLETE RESULTS FROM SECTION 5.1

Figure A1 presents the full experimental results for the 19 LLM datasets and Figure A2 presents the full experimental results for 9 weak supervision datasets in Section 5.2.

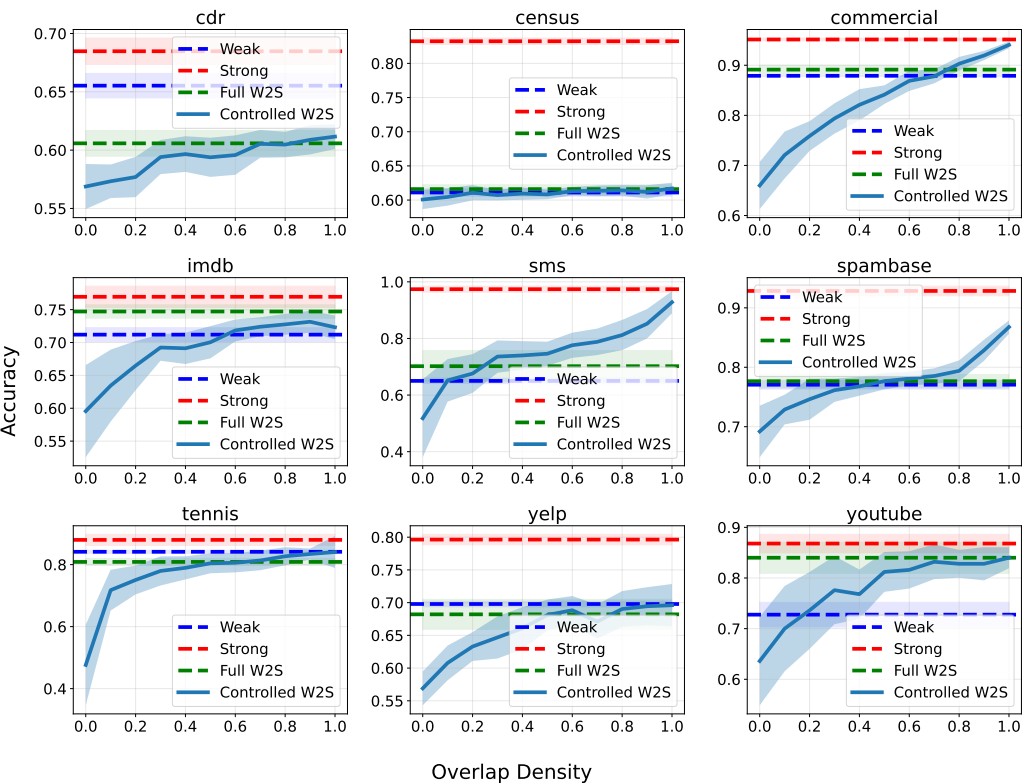

Figure A2: Overlap density mechanism in weak supervision. The red lines represent the accuracies of strong models trained on true labels, while the blue dashed lines indicate the accuracies of weak models on the test set. The green dashed lines (Full W2S) show the accuracies of weak-to-strong models trained on the entire pseudolabeled dataset. Lastly, the Controlled W2S lines represent the accuracies of strong models trained on data with a controlled proportion of overlap density. In many tasks, the strong model (a 4-layer MLP) surpasses the accuracy of the weak model (the label model) as the overlap density ratio increases. Notably, the Controlled W2S model uses less data compared to the Full WS setting to manage the overlap density proportion.

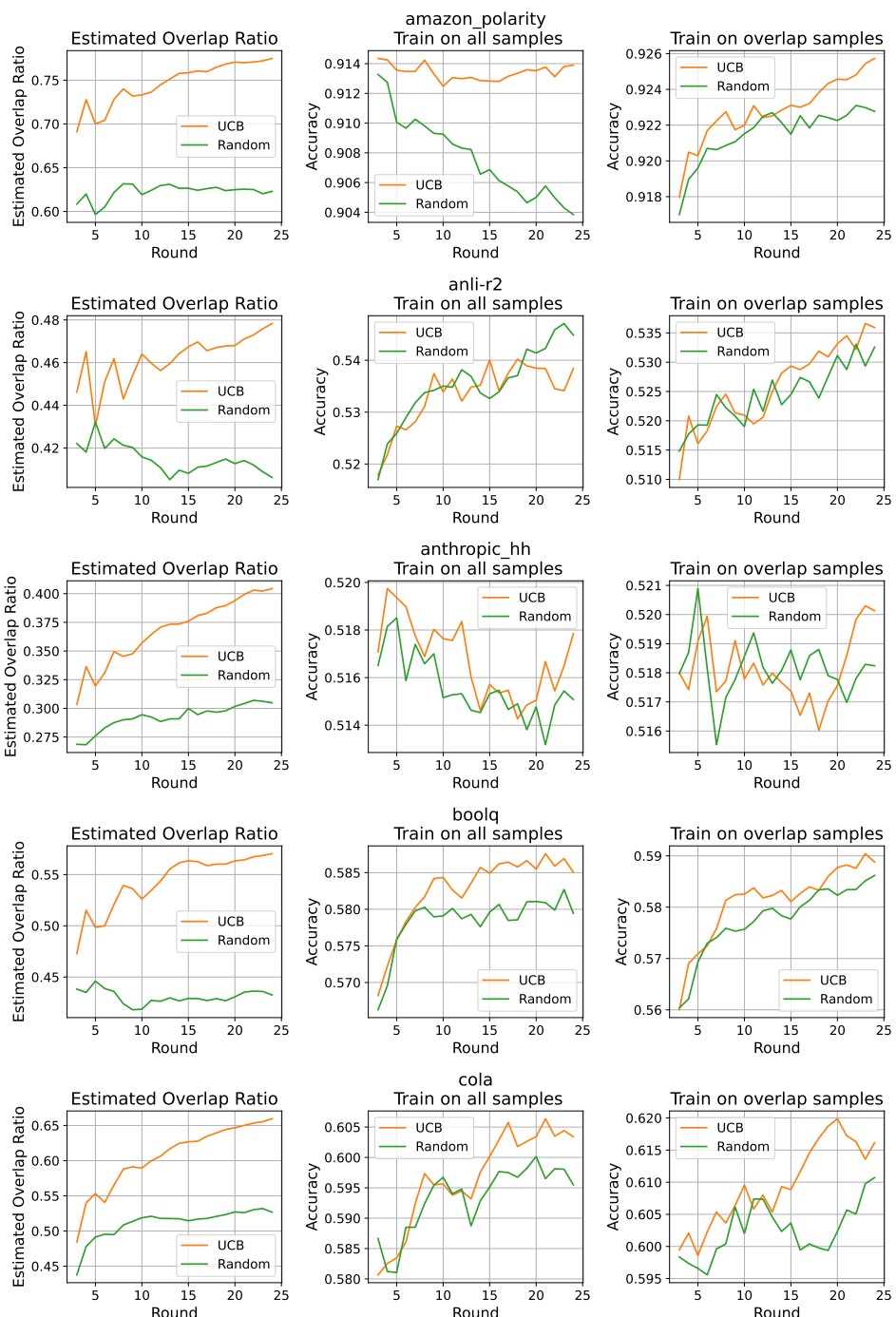

Figure A3: Data selection experiments with Algorithm 1.

## F.2    COMPLETE RESULTS FROM SECTION 5.2

Figures A3, A4, A5, and A6 present the full experimental results for the 19 datasets discussed in Section 5.2.

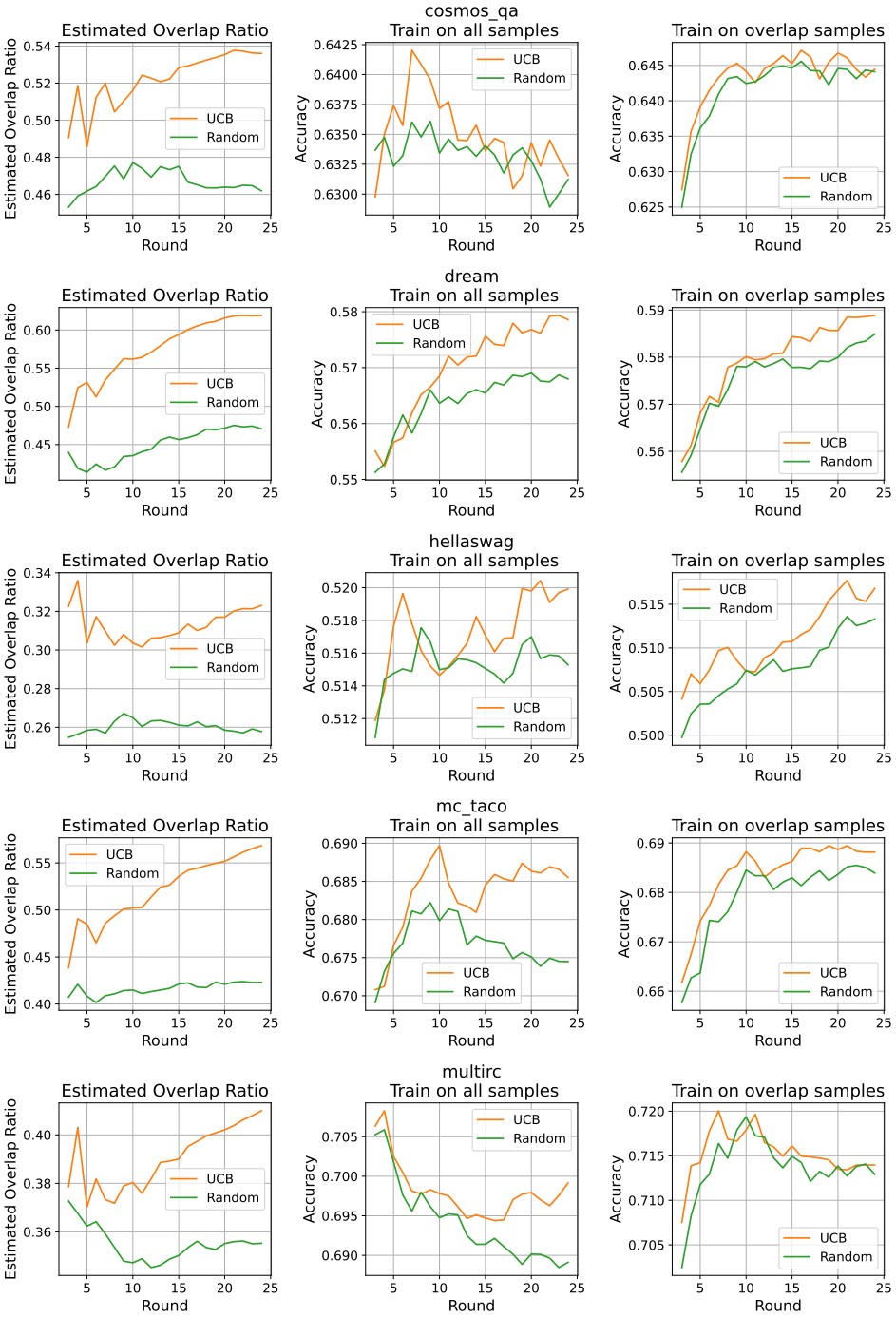

Figure A4: Data selection experiments with Algorithm 1 (Continued).

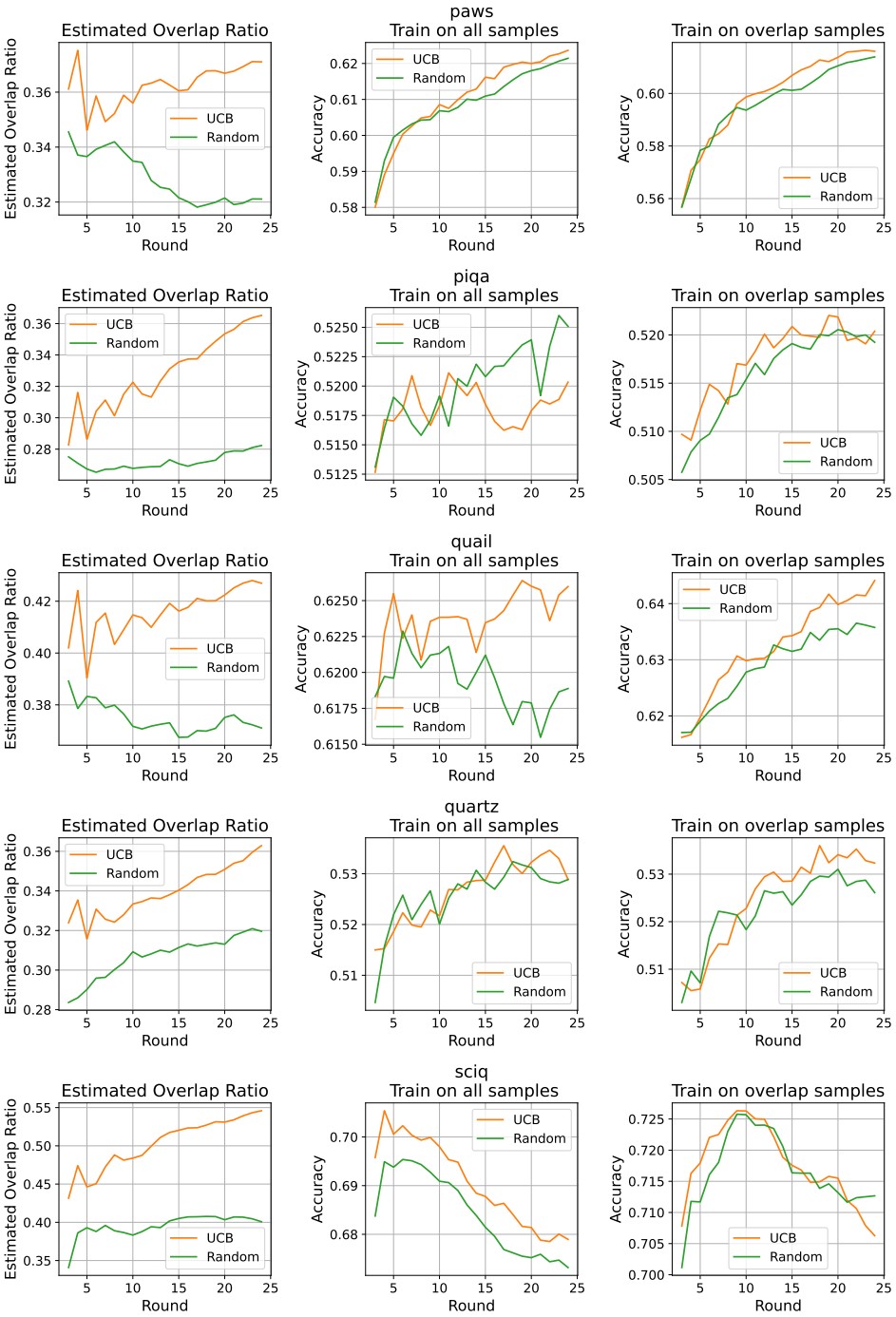

Figure A5: Data selection experiments with Algorithm 1 (Continued).

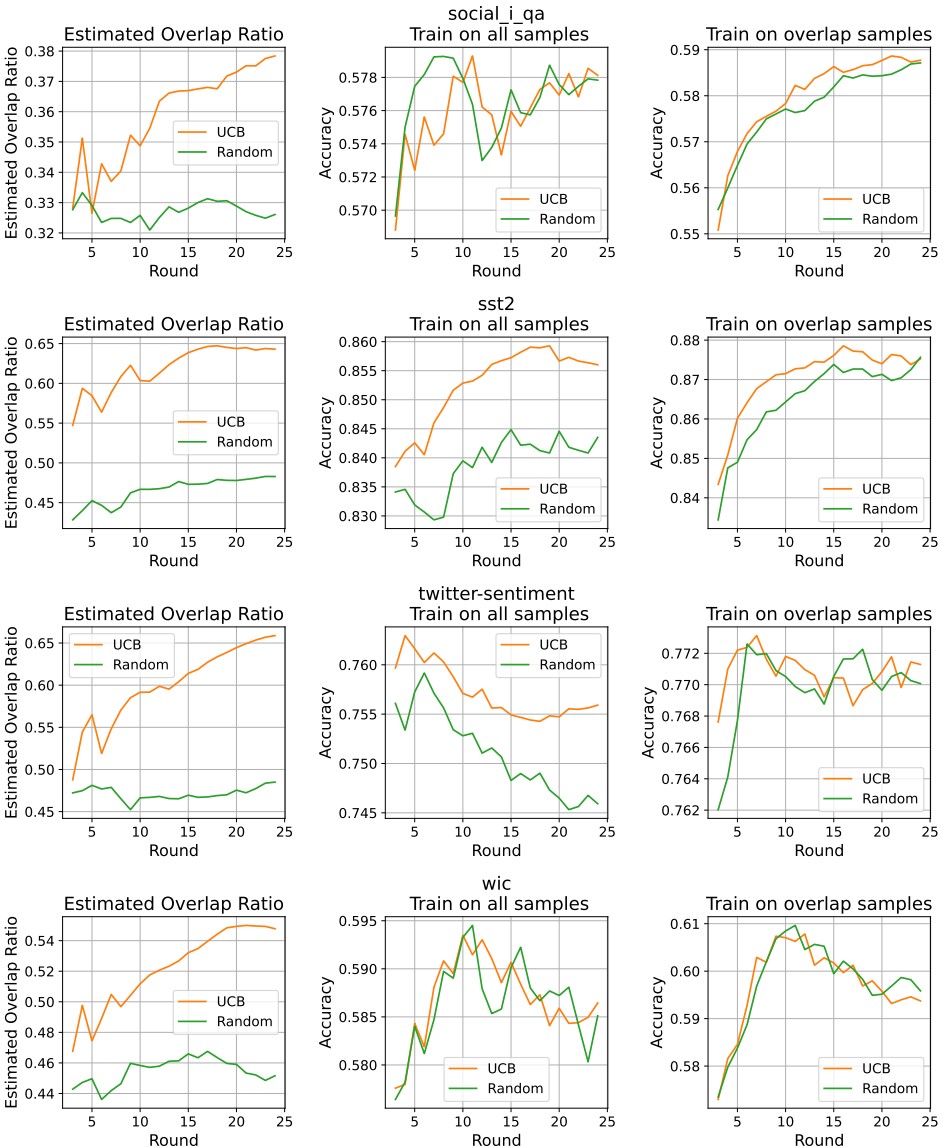

Figure A6: Data selection experiments with Algorithm 1 (Continued).

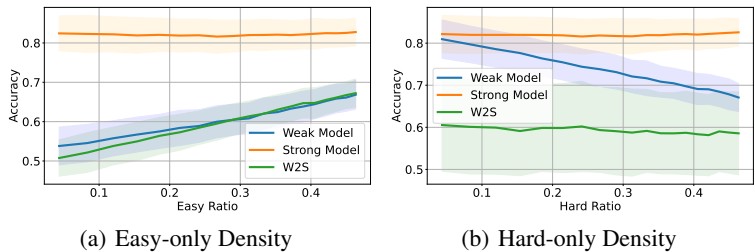

Figure A7: Ablation on the easy-only and hard-only density. Note that the y-axis represents the average accuracy across the easy, hard, and overlap data points. As expected, increasing easy-only and hard-only data points does not lead to weak-to-strong generalization.

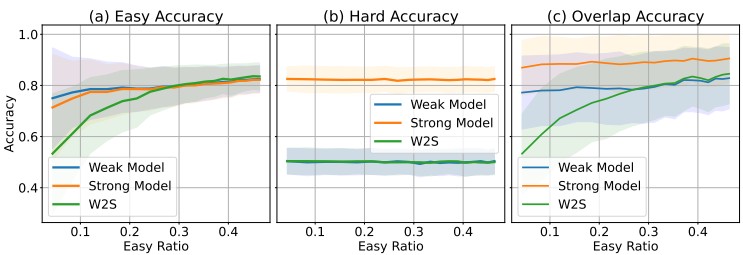

Figure A8: Accuracy in each data region in easy-only data controlled synthetic experiments. As expected, increasing the number of easy-only data points does not improve the accuracy of W2S model in hard-only data points (middle).

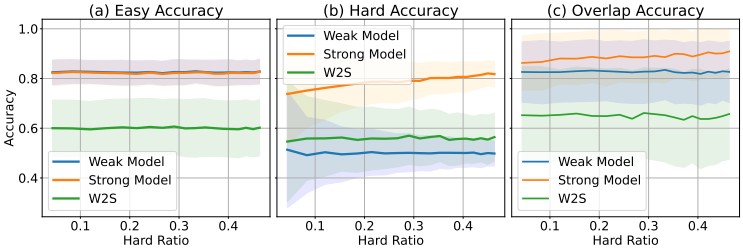

Figure A9: Accuracy in each data region in hard-onlydata controlled synthetic experiments. As expected, increasing the number of hard-only data points does not improve the performance of the W2S model due to the high error rate of pseudolabels in hard-only data points.

### F.3 SYNTHETIC EXPERIMENT ON EASY-ONLY AND HARD-ONLY DENSITY

To demonstrate that overlap density is essential for weak-to-strong generalization, we perform an ablation study focusing on easy-only and hard-only densities. We hypothesize that easy-only and hard-only points do not lead to weak-to-strong generalization.

**Setup.** We follow the experimental setup described in Section 5.3.1, with modifications to the number of easy-only, hard-only, and overlap data points. In the easy-only ratio ablation, we train the weak-to-strong model exclusively on easy-only points. We fix the number of hard-only points at 100 and overlap points at 10, then incrementally add 5 easy-only points at each step. Similarly, in the hard-only ratio ablation, we train the model exclusively on hard-only points, with the number of easy-only points fixed at 100 and overlap points at 10, while adding 5 hard-only points at each step.

**Results.** Figure A7 presents the average accuracy results in ablation experiments, and Figure A8, A9 show the decomposed views of the accuracy. The results indicate that increasing the number of easy-only points fails to achieve weak-to-strong generalization, as these points do not contribute

information to the hard-only data region, as shown in Figure A8 (middle). Meanwhile, increasing the number of hard-only points also fails, as the severe label noise impairs the learning of the weak-to-strong model.

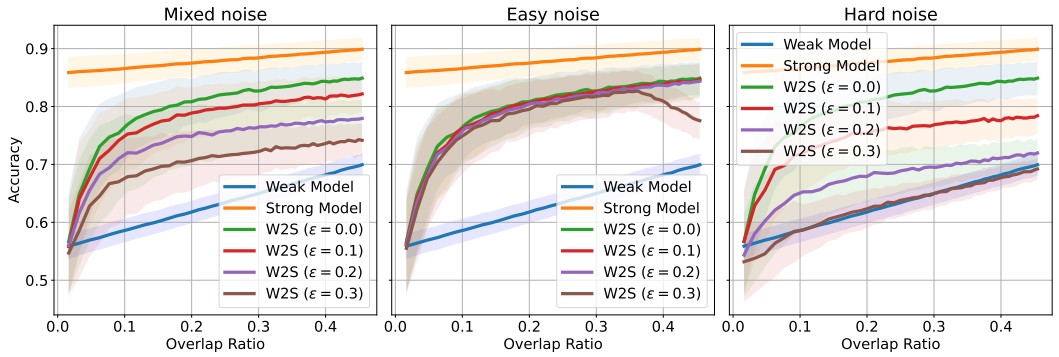

Figure A10: Average accuracy for each noise type.

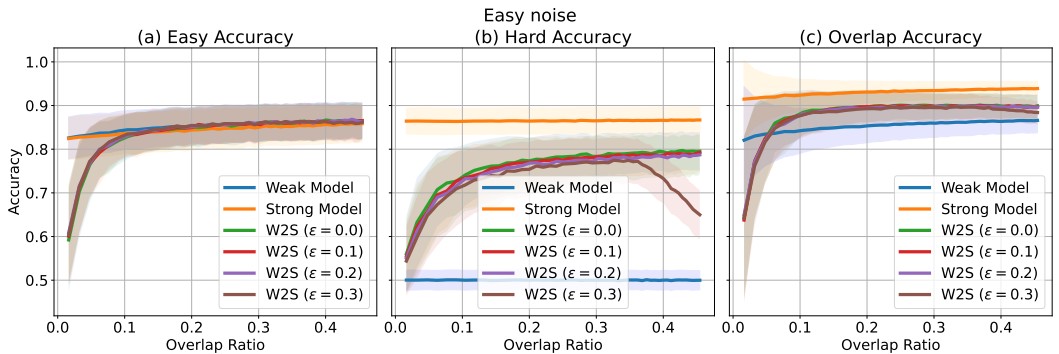

Figure A11: Accuracy in each data region for the easy noise type (N1).

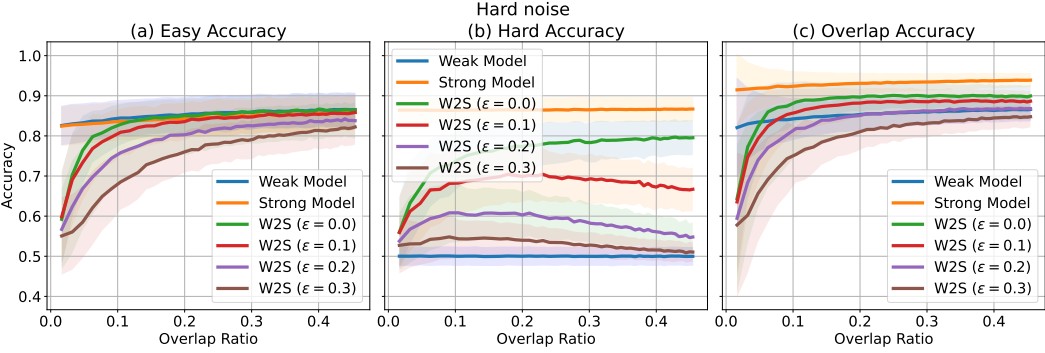

Figure A12: Accuracy in each data region for the hard noise type (N2).

## F.4 SYNTHETIC EXPERIMENT ON THE NOISE IN OVERLAP DETECTION

In the main synthetic experiment in Section 5.3, we trained a weak-to-strong model exclusively on overlap data points, under the assumption that our overlap detection algorithm is perfect. However, in practice, the overlap detection algorithm may introduce noise. In this section, we investigate the impact of noisy overlap detection on model performance in a fully controllabel experiment setup.

**Setup.** The experimental setup follows the description in Section 5.3.1 , with specific modifications to the weak-to-strong model's training data points. We examine three noise scenarios characterized

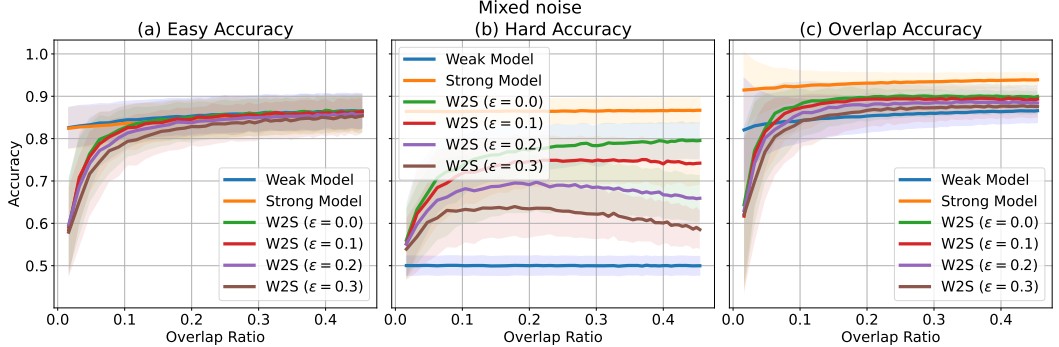

Figure A13: Accuracy in each data region for the hard noise type (N3).

by the overlap detection error rate $\epsilon$. These scenarios are as follows: (1) Mixed noise: Half of the errors select easy-only points, and the other half select hard-only points; (2) Easy noise: All errors select easy-only points; (3) Hard noise: All errors select hard-only points. The number of data points in the weak-to-strong model training set is similarly distributed as before. Starting with $n_{\text{easy}} = 100, n_{\text{hard}} = 500, n_{\text{overlap}} = 10$, we increment $n_{\text{overlap}}$ by 10 each time. Accordingly, the weak-to-strong model's training data is derived from the following distributions:

- (N1) Easy noise: $\epsilon n_{\text{overlap}}$ easy points $+ (1 - \epsilon) n_{\text{overlap}}$ overlap points

- (N2) Hard noise: $\epsilon n_{\text{overlap}}$ hard points $+ (1 - \epsilon) n_{\text{overlap}}$ overlap points

- (N3) Mixed noise: $\dfrac{\epsilon n_{\text{overlap}}}{2}$ easy points $+ \dfrac{\epsilon n_{\text{overlap}}}{2}$ hard points $+ (1 - \epsilon)$ overlap points

**Results.** Figure A10 presents the overall accuracy for each noise type, while Figure A11, A12, A13 show the decomposed accuracy for each noise type. We can observe Mixed noise and hard noise deteriorates data efficiency on overlap ratio as expected. Hard noise, purely adding randomly labeled hard points, significantly drops data efficiency of overlap data points. Interestingly, while easy noise has minimal impact at low error rates, it significantly degrades performance when the error rate is high ($\epsilon \geq 0.3$). This degradation is due to the model assigning higher weights to easy features under high easy noise conditions, leading the weak-to-strong model to over-rely on these features at high error rates.

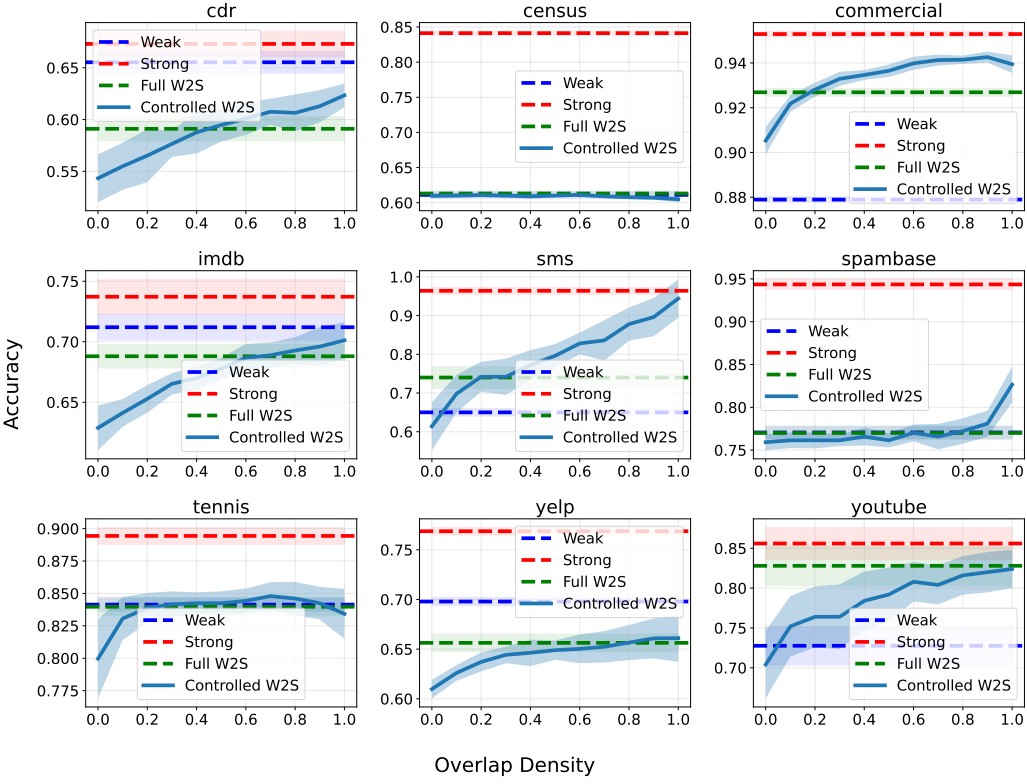

Figure A14: Overlap density mechanism in weak supervision with XGBoost as the weak-to-strong model. The red lines represent the accuracies of strong models trained on true labels, while the blue dashed lines indicate the accuracies of weak models on the test set. The green dashed lines (Full W2S) show the accuracies of weak-to-strong models trained on the entire pseudolabeled dataset. Lastly, the Controlled W2S lines represent the accuracies of strong models trained on data with a controlled proportion of overlap density.

## F.5 OVERLAP DENSITY MECHANISM WITHOUT NEURAL NETWORKS

One might assume that the overlap density mechanism is primarily a feature of deep neural network architectures, given that our experiments mainly use deep neural networks as strong models. While deep neural networks offer useful representations, we demonstrate that the overlap density mechanism can also be shown even without the use of neural networks.

**Setup.** We adopt the same weak supervision experiment setup as in Section 5.1, except that we use raw inputs for the overlap detection algorithm and XGBoost (Chen & Guestrin, 2016) as the strong model.

**Results.** Figure A14 presents the experimental results. Although the outcomes appear noisier due to the less powerful representations, which lead to a noisier overlap detection algorithm, we can still observe that the overlap density mechanism is effective—improvements in the weak-to-strong model correspond with increases in overlap density.

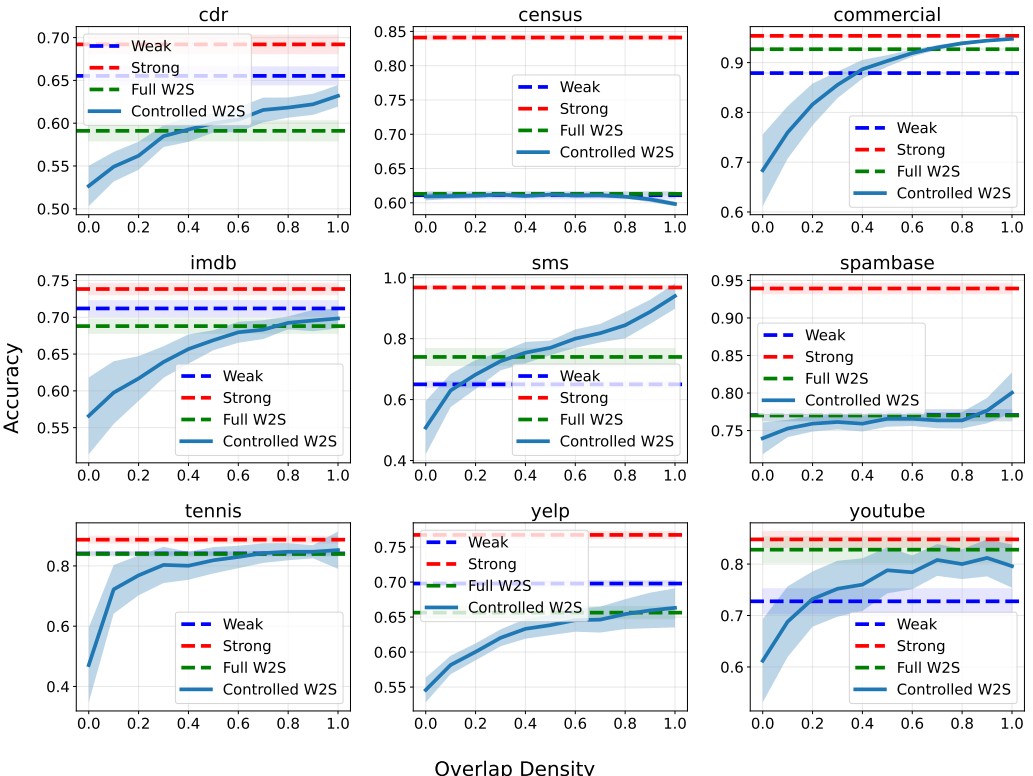

Figure A15: Overlap density mechanism in weak supervision with XGBoost as the weak-to-strong model and transferred overlap points from 4-layer ReLU networks. The red lines represent the accuracies of strong models trained on true labels, while the blue dashed lines indicate the accuracies of weak models on the test set. The green dashed lines (Full W2S) show the accuracies of weak-to-strong models trained on the entire pseudolabeled dataset. Lastly, the Controlled W2S lines represent the accuracies of strong models trained on data with a controlled proportion of overlap density.

## F.6 TRANSFERABILITY OF DETECTED OVERLAP DENSITY

Since overlap detection relies on the model's representation, one might assume that overlap points are model-dependent — different weak models and weak-to-strong models would have different overlap points. However, we hypothesize that the overlap property is a latent property of data and therefore the detected overlap points are transferable across models.

**Setup.** We use the same setup as in Section 5.1, except that overlap/non-overlap points are detected using a 4-layer DNN trained on pseudolabels in $D_{\text{w2s}}$. and then the weak-to-strong model evaluation is performed with XGBoost after training it on $D_{\text{w2s}}$.

**Results.** Figure A15 shows the experimental results. We can observe a similar trend to that in Section 5.1, supporting our hypothesis.

