# OpenReview forum: "Weak-to-Strong Generalization Through the Data-Centric Lens"
_ICLR.cc/2025/Conference — ICLR 2025 Poster_

### Official Review · Reviewer_3AtN · 2024-10-26

**Soundness:** 3
**Presentation:** 3
**Contribution:** 3
**Rating:** 6
**Confidence:** 3

**Summary:**

The weak-to-strong generalization in this paper refers to the phenomenon that strong model can generalize better on the unseen data (or say hard patterns) than the weak model, even only being trained on supervision (e.g. pseudolabel) obtained from weak model. To solve the challenge, this paper proposes a data-centric mechanism, named overlap density, to characterize weak-to-strong generalization ability, provide an overlap detection algorithm to find the points with high overlap density from one data source, and then leverage them to facilitate learning among multiple source of data. This paper further provides theoretical analysis and empirical experiments to demonstrate the effectiveness of the proposed overlap density and overlap detection algorithm.

**Strengths:**

(1)	This paper provides a novel and practical data-centric mechanism, overlap density, to measure what aspects of data enable weak-to-strong empirical performance.

(2)	The proposed overlap detection algorithm (i.e. Algorithm 2) is new for me, and particularly, is simple and practical in real-life applications. Besides, using UCB algorithm to choose the data source with the highest overlap density is also of high interest.

(3)	The proposed error bound in Theorem 4.1 directly reveals the influence of overlap density and error rate in estimating the correct overlap density to the weak-to-strong generalization ability of strong model.

(4)	The concentration inequality in Theorem 4.2 theoretically demonstrates with high probability, the Algorithm 2 (i.e. overlap detection algorithm) can select truly overlap dataset, which truly makes sense. The regret bound in theorem 4.3 also guarantees the asymptotic performance of Algorithm 1.

(5)	The experiments are also extensive, and in my opinion truly validate the effectiveness of the proposed mechanism from an empirical view.

Overall, I find this paper is novel. The proposed overlap density mechanism, the overlap detection algorithm, the theoretical guarantees for the proposed mechanism and algorithms, as well as the extensive experiments, form a full story. I check the proof as possible as I can, and I think it is sound.

**Weaknesses:**

(1)	The major weakness is that many important notations, in both the main paper and the appendix, lack clear explanations. For example, the most important concept proposed in this work is the so-called overlap density, then where is the formal definition (or say mathematical definition) of it? I could not find such rigorous definition until I read line 16 in Algorithm A1 in appendix. I believe the formal definition of such mechanism should be depicted very clearly in the main paper. Besides, in pseudo code of Algorithm 1, the upper confidence bound of overlap density should be explained clearly in both Algorithm 1 and the main paper, instead of deferring it into Algorithm A1, because it is very important in UCB algorithm and truly help readers to understand what pseudocode means.

(2)	If I understand correctly, the confidence scores in step 5 of Algorithm 2 lack clear explanations either. There is no explanations for how to compute it in the main paper.


(3)	The adversarial robustness assumption mentioned in Theorem 4.1 should at best be listed in the main paper (although its definition is a little lengthy), or at least should be clearly pointed out in which part of Appendix such definition locates.

(4)	The mathematic definition of the empirical overlap ratio and optimal overlap density should be given before Theorem 4.3.


(5)	in Definition 1, what does $\mathcal{N}$ mean? Neighbor? Should be explained.

(6)	In Theorem 4.3, the regret bound is of $O(\sqrt{K\log{T}/t})$, not the common forms of $O(\sqrt{K\log{t}/t})$. This means that Algorithm 1 could not achieve a vanishing regret in the first $\log{T}$ rounds. What is the insightful reason for it? Can we improve this regret upper bound?

Minor:

(7)	Line 21, leverage it -> leverage them

(8)	Line 22 densityand

(9)	Line 154, Dw2s

(10)	Line 224 in Algorithm 2, $\tau$ -> $\tau _{hard}$

(11)	Line 295, addiitional

(12)	Line 846, WWe

(13)	Line 854, is $f _{weak}(y)(x)$ a typo?

(14)	Line 1097, lack left parenthesis.

(15)	Line 1220, $o(s)$ lacks formal definition.

(16)	Line 1248, $R(t,a)$ -> $R(t,s)$

**Questions:**

(1)	Can you give more theoretical explanations for why the regret bound in Theorem 4.3 is $O(\sqrt{\log{T}/t})$, not $O(\sqrt{\log{t}/t})$?

---

### Official Review · Reviewer_enrk · 2024-10-29

**Soundness:** 3
**Presentation:** 4
**Contribution:** 3
**Rating:** 8
**Confidence:** 2

**Summary:**

This work focuses on the weak-to-strong (W2S) generalisation mechanism, focusing on making an agent with strong generalisation ability to learn under the supervision of a weak agent, a paradigm of interest, eg, in the context of superalignment for LLMs. While W2S generalisation is already well studied from an algorithmic perspective, little work has been made on the intrinsic structures in data that allow W2S generalisation to be made . This work is then a step in this direction, authors propose and discuss the notion of $\emph{overlap}$ which consists of data containing both weak patterns (usable by the weak agent) and strong ones (only exploitable by the strong agent). Such a structure is meaningful as those points allow the strong agent to learn from the weak ones and beyond. Authors propose algorithms to characterise, recognise and select overlapping data. Then, then propose a short theoretical analysis of their results and show that real-life dataset may contain overlapping data and that those data may explain the W2S generalisation phenomenon.

**Strengths:**

- The paper is well-written, as a non-expert of W2S generalisation, goals are clearly stated and the notion of overlap, while hand-wavy, is well explained.
- The empirical study is very complete and shows both relevance and limitation of the current approach of overlapping with only two distinguishable patterns.
- Theoretical result are various and justify why overlapping and the proposed algorithms should be considered.

**Weaknesses:**

See Questions

**Questions:**

- l.155 $f_{strong}$ is not defined, do you mean $f_{w2s}$?
- About theorem 4.1: can we reasonably hope for $c$ to be greater than $0.5$ in practice? Is there a way to estimate it? It seems that this condition is the key to kill the impact of  $\varepsilon_2$ . More generally is there a way to detect whether a data distribution satisfies a $(c,q)$ expansion?
- As acknowledged by authors, the data model $x= [x_{easy},x_{hard}]$ is unlikely to be verified in practice, what is the justification to focus on such a particular case then?
- Although it is not about weak to strong generalisation, the idea of selecting data wrt relevant information already appeared recently in statistical learning. In [1,2], authors derived the Pick 2 Learn (P2L) algorithm, which compress the available dataset in order to maintain the major part of relevant information while strongly decreasing the dataset size. I suggest you to briefly discuss this to provide more perspective in your work.
- Is it possible to use simply algorithm 2 in a learning setting where you have only 1 dataset? Does it lead to good performance?
- In Algorithm 2, you fix your hard and overlap threshold yourself. How do you choose them?
- At the beginning of page 4 I remark that having two patterns (weak and strong) leads to 4 subsets to partition data, if we add multiple level of overlapping as suggested in your conclusion, this may result in a exploding number of subsets to partition data.  Would it be a problem to adapt both the algorithms and theoretical results?

Biblio:
[1] Compression, generalization and learning. Campi et al. 2023, Journal of Machine Learning Research
[2] The Pick-to-Learn Algorithm: Empowering Compression for Tight Generalization Bounds and Improved Post-training Performance, Paccagnan et al. NeuriPS 2023

---

### Official Review · Reviewer_5BLm · 2024-11-04

**Soundness:** 4
**Presentation:** 2
**Contribution:** 4
**Rating:** 6
**Confidence:** 3

**Summary:**

The paper provides a notion of data overlap to describe the process of weak-to-strong generalization, which occurs when one uses pseudo labels from a “weak” model to train a “strong” model to achieve better performance. The paper presents a theoretical analysis of the error of the strong model and provides experiments on both language modeling tasks and weak supervision tasks.

**Strengths:**

The theoretical results seem correct to me (though I did not go through the appendix), and the experiments are comprehensive, illustrating that the results described in the paper also hold in practice.

**Weaknesses:**

My main concern is the clarity of the writing. I found it hard to understand the paper when reading it, due to the use of some terms without proper definitions (see questions).

**Questions:**

1. The paper talks about focusing on improving generalization but does not clearly define what this means. In line 88, the paper mentions, “we seek to understand what aspects of data result in stronger performance,” but it does not define what “performance” means here. In line 141, the paper mentions, “how can we prioritize sources that lead to the greatest generalization?”

   Are we measuring performance on test points drawn from the same distribution as the training data, or something else? It would be helpful to define “performance” or “generalization” clearly early on. Furthermore, the paper discusses using overlap detection to improve learning from multiple datasets, making the notion of performance even less clear.

2. The terms “easy pattern” and “hard pattern” are rather informal and somewhat vague. The assumptions about easy and hard patterns, which force $x_{\text{easy}}$ and $x_{\text{hard}}$ to be zero in lines 177–182, seem very strong.

3. Typo in line 154: “Dw2s” , w2s is in the subscript ?

4. Some important assumptions are not clearly stated in the paper and are instead embedded within paragraphs. For example, the assumption in lines 242–250 seems very important.

   It would be helpful if the paper could reorganize and clarify what crucial assumptions are needed.

5. Typo in line 290: “$\text{err}(f_{w2s}, y \dots)$” to “$\text{err}(f_{w2s}, f_{\text{weak}} \dots)$” ?

6. Is there any experiment that verifies that the strong model only learns hard patterns?

---

### Official Review · Reviewer_YoBV · 2024-11-06

**Soundness:** 3
**Presentation:** 3
**Contribution:** 3
**Rating:** 6
**Confidence:** 3

**Summary:**

The paper discusses weak-to-strong generalization, the process by which a large capacity model is trained using pseudo-labels generated by the predictions of a smaller model.
The goal is to have the final model achieve better performance than what would have been obtained by simply training on the available ground truth labels, or by the weaker models.

A hypothesis is raised regarding the mechanism that facilitates this type of generalization. Namely, it is hypothesized that there are two types of patterns associated with the label, "easy" ones that the smaller model can learn and "hard" ones that it cannot.
The hypothesis is then that examples in the dataset which contain both easy and hard patterns, form an overlap between the examples that the weak model can classify successfully and those that the stronger model can classify using the hard patterns.
Pseudo-labels for this overlap set then help the larger model leverage the hard patterns learn them and classify examples that do not contain easy patterns.

Based on this hypothesis, a method for selection of data for pseudo-labelling is proposed, and some formal results are given to show upper bounds on error that depend on the amount of overlap, and for motivating the proposed sample selection methods using a toy problem with Gaussian data. In experiments, the improvements due to larger overlap and how the sample selection method can be used to induce larger overlap, are demonstrated by linear probing of LLMs on a variety of datasets, and on the proposed toy problem.

**Strengths:**

The paper touches upon an important problem and raises an original, interesting and intuitive hypothesis. The writing is conveys the contributions of the paper clearly, and the findings seem significant, even though I am not an expert on this topic so I am not entirely sure what are the appropriate baselines.
Finally, I also like that the authors reasoned formally about the empirical phenomenon and method.

**Weaknesses:**

I think that the paper covers a lot of material at the expense of some details that would've made the argument more precise and convincing.

1) The result in Theorem 4.1 makes sense in terms of being an upper bound on the error we are concerned with, but I could not see immediately whether the bound is very loose or not. For instance, it looks like if $f_{weak}$ preforms poorly on on $S_i \cap D_{hard only}$, then an accurate $f_{w2s}$ means the term on the RHS should be very large (since the disagreement with $f_{weak}$ on this set is large) and the bound becomes vacuous. A better explanation for the significance of this bound would be much appreciated. The example given for a scenario where the bound reduces to the disagreement between $f_{w2s}$ and $f_{weak}$ on the hard-only examples is nice, but still the right hand side can be very large and in we also expect it to be large when $f_{w2s}$ gives a significant improvement. Hence, seems like this bound also has the property of becoming loose when the error of $f_{w2s}$ becomes smaller.

Since the result here relies on a paper from Lang et al. 2024 that is an arXiv paper and readers in ICLR may have not had the time to read and appreciate yet, I think that such clarifications are even more important.

2) It is unclear how the hyperparameters $\tau_{\text{hard}}$ and $\tau_{\text{overlap}}$ are selected.

3) For the experiments, I did not see why the results convey that the improved performance indeed comes from succeeding on "hard-only" examples, and not just from better generalization to overlap examples. A study that shows this more fine-grained analysis, and how accuracy depends on $\tau_{\text{hard}}$ and $\tau_{\text{overlap}}$ would be appreciated.

**Questions:**

In figure 2, is there a reason that different datasets are presented for each regime (low, medium and high), instead of showing 2 datasets through all regimes? Since the amount of overlap is induced by selection, I'd imagine this would give a more apples-to-apples comparison of the regimes.

Minor comments:
In definition 1, what does $\mathcal{N}(U)$ denote? Is it simple supposed to be $U$?
Line 278: information -> informal
Lines 248-249: "while easy-only points lack are not"
Line 312: what's $\tilde{x}$?

---

### Author Response · Authors · 2024-12-03
**Discussion Summary**

Dear Reviewers/AC/SAC/PC,

We sincerely thank you for your valuable feedback and constructive suggestions. Below, we summarize the major revisions made during the rebuttal process:

* **Presentation**: To enhance clarity and readability, we have moved key content from the appendix to the main body of the paper. In particular, we clarified previously missing definitions and provided additional context and explanations to improve the flow and coherence of the paper.
* **Typos and Errors**: We carefully reviewed and corrected all typos and minor errors highlighted by the reviewers.

Thank you once again for your insightful comments and the opportunity to improve our work.


Best regards,

The Authors

---

### Meta-Review · Area_Chair_Qdc4 · 2024-12-10

**Metareview:**

This paper presents a new notion termed overlap density that serves as a metric for weak-to-strong generalization. Reviewers found that the metric is well motivated, and the algorithm exploring to optimize this metric is easy to follow. Authors also made efforts to provide generalization bounds based on certain assumptions. On the other side, reviewers found that the main theorem may be vacuous in certain regimes, and that the connection and distinction to a recent work of Lang et al 2024 were not clearly stated.

After author rebuttal and AC-reviewer discussion, reviewers agreed that the strengths outweigh weaknesses.

**Additional Comments On Reviewer Discussion:**

Authors addressed many clarity and technical questions. However, there is still concern on the main theorem. In the end, reviewers feel that there are novel aspects and thus this work is above the bar.

---

### Decision · Program_Chairs · 2025-01-22

Accept (Poster)